# A genome-phenome association study in native microbiomes identifies a mechanism for cytosine modification in DNA and RNA

Weiwei Yang[1†], Yu-Cheng Lin[1,2†], William Johnson[1‡], Nan Dai[1], Romualdas Vaisvila[1], Peter Weigele[1], Yan-Jiun Lee[1], Ivan R Corrêa Jr[1], Ira Schildkraut[1], Laurence Ettwiller[1]*

[1]New England Biolabs, Ipswich, United States; [2]School of Dentistry, National Yang Ming Chiao Tung University, Taipei, Taiwan

**Abstract** Shotgun metagenomic sequencing is a powerful approach to study microbiomes in an unbiased manner and of increasing relevance for identifying novel enzymatic functions. However, the potential of metagenomics to relate from microbiome composition to function has thus far been underutilized. Here, we introduce the Metagenomics Genome-Phenome Association (MetaGPA) study framework, which allows linking genetic information in metagenomes with a dedicated functional phenotype. We applied MetaGPA to identify enzymes associated with cytosine modifications in environmental samples. From the 2365 genes that met our significance criteria, we confirm known pathways for cytosine modifications and proposed novel cytosine-modifying mechanisms. Specifically, we characterized and identified a novel nucleic acid-modifying enzyme, 5-hydroxymethylcytosine carbamoyltransferase, that catalyzes the formation of a previously unknown cytosine modification, 5-carbamoyloxymethylcytosine, in DNA and RNA. Our work introduces MetaGPA as a novel and versatile tool for advancing functional metagenomics.

## Editor's evaluation

This work will interest researchers who want to explore the functional potential of metagenomes. The authors present an original approach, MetaGPA, for performing enrichment analysis on cohorts of metagenomes and use it to identify a novel enzyme that can modify cytosines in DNA from natural bacteriophage populations.

*For correspondence:
laurence.ettwiller@gmail.com

†These authors contributed equally to this work

Present address: ‡Department of Molecular Biology and Microbiology, Tufts University, School of Medicine, Boston, United States

## Introduction

Advances in next-generation sequencing technology have been reshaping metagenomics, making it possible to explore all microbes within a sample, including the overwhelming majority of those unculturable in standard growth conditions (*Quince et al., 2017*). From studying human gut microbiome to marine viral communities (*Sunagawa et al., 2015*), metagenomic analyses are being utilized across a broad spectrum of life science disciplines, contributing to novel clinical diagnoses (*Chiu and Miller, 2019*), antibiotics and small molecule discovery (*Hu et al., 2013*; *Charlop-Powers et al., 2014*), food safety (*Cao et al., 2017*), biofuel generation (*Hess et al., 2011*), and environment stewardship (*Nesme et al., 2014*; *Cavicchioli et al., 2019*).

Of the two primary questions metagenomic studies strive to address, profiling the taxonomic biodiversity – 'what is out there?' is easier to answer compared to inferring the biological functions

**eLife digest** Many industrial processes, such as starch processing and oil refinement, use chemicals that cause harm to the environment. These can often be switched to more sustainable biological processes that are powered by proteins called enzymes.

Enzymes are micro-factories that speed up biochemical reactions in most living things. Communities of microorganisms (also known as microbiomes) are an amazing but often untapped resource for discovering enzymes that can be harnessed for industrial purposes. To gain a better picture of the microbes present within a population, researchers often extract and sequence the genetic material of all microorganisms in an environmental sample, also known as the metagenome. While current methods for analyzing the metagenome are good at identifying new species, they often provide limited information about the microorganism's functional role within the community. This makes it difficult to find new enzymes that may be useful for industry.

Here, Yang, Lin et al. have developed a new technique called Metagenomics Genome-Phenome Association, or MetaGPA for short. The method works in a similar way to genome-wide association studies (GWAS) which are used to identify genes involved in human disease. However, instead of disease associated genes in humans, MetaGPA finds microbial genes that are associated with a biological process useful for biotechnology.

Like GWAS, the new approach created by Yang, Lin et al. compares two groups: the first contains microorganisms that carry out a specific process, and the second contains all organisms in the microbiome. The metagenome of each group is extracted and a computational pipeline is then applied to identify genes, including those coding for enzymes, that are found more often in the group performing the desired task.

To test the technique, Yang, Lin et al. used MetGPA to find new enzymes involved in DNA modification. Microbiome samples were collected from coastal water and sewage, and the computational pipeline was applied to discover genes that are associated with this process. Further analysis revealed that one of the identified genes codes for an enzyme that introduces a previously unknown change to DNA.

MetaGPA could be applied to other processes and microbiomes, and, if successful, may help researchers to identify more diverse enzymes than is currently available. This could scale up the discovery of new enzymes that can be used to power industrial reactions.

– 'what do they do?'. Indeed, there have been many well-established strategies to quantify taxonomic diversity such as analyzing known marker genes, binning contigs and assembling sequences into taxonomic groups or genomes (*Sharpton, 2014*; *Quince et al., 2017*). However, a large fraction of the functional potential of microbiomes remains to be discovered. Microbiomes encode a large number of evolutionarily diverse genes coding for millions of peptides and proteins for distinctive functions. The functional diversity is particularly enormous for secondary metabolites and epigenetic modifications (*Sharon et al., 2014*). Taking DNA modifications as an example , environmental microbiomes are a plentiful source of DNA-modifying enzymes with diverse mechanisms (*Weigele and Raleigh, 2016*; *Hiraoka et al., 2019*). In bacteriophages, DNA modifications are used to evade the restriction modification systems of the host bacterial cell. Notably, a handful of bacteriophages have been described to completely modify cytosines in their genomic DNA, for example, 5' methylcytosine in XP12 phage (*Kuo et al., 1968*) and glycosylated 5' hydroxymethylcytosine in T4 phage (*Revel and Georgopoulos, 1969*). Recently, additional base modifications have been discovered including 5-(2-aminoethoxy) methyluridine, 5-(2-aminoethyl)uridine, and 7-deazaguanine (*Lee et al., 2018*; *Hutinet et al., 2019*).

Current methods to harness such functional potential of microbiomes, for the most part, came from computational prediction of gene function which is based on homology search to existing databases. Nonetheless, because of the poor completeness and accuracy of microbial annotation, homology searches often fail to impute the correct functions (*Huson et al., 2009*; *Nayfach and Pollard, 2016*). Novel functions can also be found using functional screens; however, such effort entails the construction and high-throughput screening of a large number of clones which can be very time-consuming (*Martinez et al., 2004*). Novel approaches are therefore needed to link genetic information from microbial metagenomes to function.

For specific bacteria, determining the genetic basis of phenotypes has been addressed using genome-wide association studies (GWAS) (*Falush and Bowden, 2006*) combined with specific phylogenetic methods to account for the unique population structure of microbes (*Collins et al., 2018*). Examples of microbial GWAS have explored hundreds of isolates to identify genomic elements that are statistically associated with, for example, antibiotic resistance (*Chewapreecha et al., 2014*), host specificity (*Sheppard et al., 2013*), or virulence (*Laabei et al., 2014*). Nonetheless, these studies are limited to known isolates and have not yet been extended to entire complex microbial communities.

Here, we developed the Metagenomics Genome-Phenome Association (MetaGPA) framework to bridge the gap between genetic information and functional phenotype in complex microbial communities. MetaGPA is conceptually close to GWAS as it associates genotypic data with phenotypic traits. Contrasting with microbial GWAS which uses sequence variations as genetic markers, MetaGPA incorporated association analyses at the level of protein domains to reveal genes that are significantly associated with the phenotype of interest. By applying this workflow on DNA modifications as our phenotypic trait, we discovered a number of candidate enzyme families. From these candidates, we validated a novel DNA/RNA cytosine modification, 5-carbamoyloxymethylcytosine (5cmdC), and the enzyme responsible for this modification. From this example, we show that MetaGPA is a powerful and versatile method to improve metagenome functional analysis.

## Results
### Conceptual framework of MetaGPA studies

Like GWAS in individual species (*Hirschhorn and Daly, 2005*), MetaGPA requires definition of two cohorts, a 'case' cohort, that is, a group of organisms that share a specific phenotype under study in a given microbiome, and a 'control' cohort composed of all organisms within that microbiome (*Figure 1* and *Figure 1—figure supplement 1*). While both cohorts are sequenced independently, all organisms included in a cohort are sequenced together without the need to isolate organisms. The 'case' group is derived from the control group after applying a selection to only retain a given phenotype.

The association is computed using a computational workflow composed of the core MetaGPA pipeline that defines genetic units associated with the phenotype and further analysis tools to refine these associations. The core MetaGPA pipeline described in *Figure 1* and *Figure 1—figure supplement 1* takes sequencing reads from both cohorts and performs de novo assemblies into contigs. Contigs from both cohorts are combined and duplicated contigs are discarded. Reads from the case and control experiments are mapped back to the combined contig set. Each contig is either labeled as enriched or depleted from the case cohort based on the enrichment score calculated using the relative number of reads mapping to the contig. Genes identified in the contig set are annotated using homology search to known protein domains and domains that are found significantly enriched in the enriched contig set are considered to be associated with the studied phenotype. Finally, genes in the enriched contig set annotated with one or more associated domain(s) are defined as candidate genes.

Both associated domains and their corresponding candidate genes are further refined using evidence such as phylogenetic clustering and co-occurrence with other candidate domain families/genes. Phylogenetic clustering assesses for each associated domain, whether candidate genes are phylogenetically closer to each other relative to genes containing the same domain family in the depleted contigs. This analysis can be done at domain or residue resolution. Co-occurrence with other candidate genes strengthen the association by attempting to identify the entire metabolic pathway responsible for the specific phenotype under study. Taken together, this multilayer analysis effectively identifies protein domain families and their corresponding candidate genes that are truly related to the phenotypes of interest.

### Application of MetaGPA for the discovery of cytosine modifying and hyper-modifying enzymes
#### Principle

To demonstrate the effectiveness of the MetaGPA framework, we designed a study to identify proteins associated with DNA cytosine modifications. Because the relevant phenotype (cytosine modifications) is covalently linked to the genetic material, total genomic DNA isolated from environmental sources can be used for both, the 'control' cohort and material for phenotypic selection.

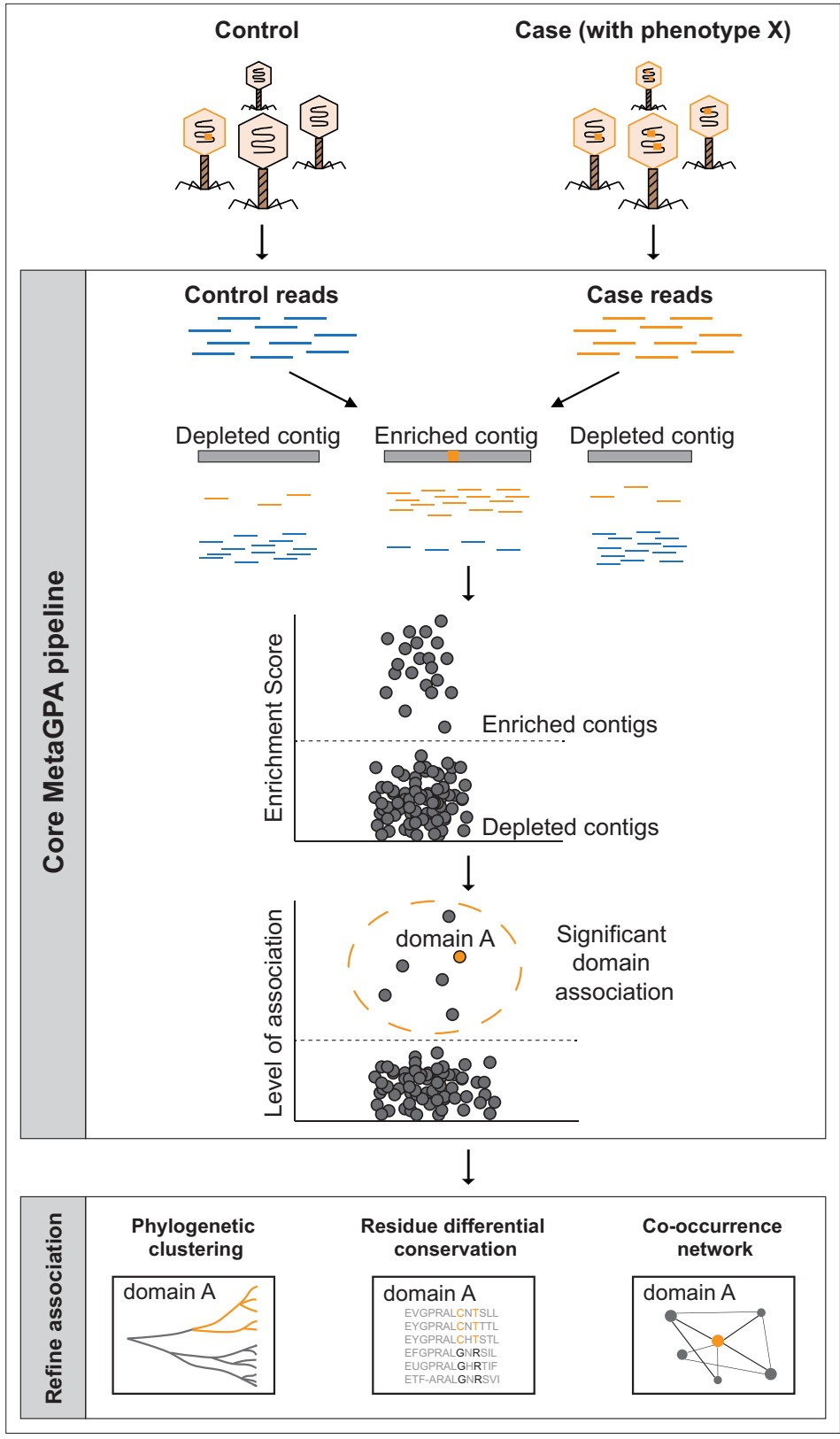

**Figure 1.** Schematic overview of Metagenomics Genome-Phenome Association (MetaGPA). The core MetaGPA pipeline is applied to identify protein domains that are significantly associated with a given phenotype. The pipeline takes high-throughput sequencing datasets from two libraries representing (1) the total microbiome (control reads) and (2) a subset of the microbiome resulting from a given treatment that selects for the studied

*Figure 1 continued on next page*

*Figure 1 continued*

phenotype (case reads). Control and case reads are assembled into contigs and contigs are classified as enriched or depleted based on the ratio between the number of reads mapping to the contigs from the case versus control. Contigs are annotated based on homology with known protein domains and each domain is evaluated for its association with the studied phenotype based on the number of hits in the enriched compared to control contigs. In this theoretical example, the orange bar represents a protein domain that is associated with phenotype X (e.g., DNA cytosine modification, see below). We then apply additional metrics such as phylogenetic clustering, differential conservation, and co-occurrence network to refine this association and identify candidate genes for experimental validation. More details about the workflow are provided in *Figure 1—figure supplement 1*.

The online version of this article includes the following figure supplement(s) for figure 1:

**Figure supplement 1.** Detailed workflow of Metagenomics Genome-Phenome Association (MetaGPA).

Accordingly, a 'case' cohort is obtained by applying an enzymatic selection to retain only the genomic DNA containing known and unknown forms of cytosine modification. More specifically, unmodified cytosines are deaminated to uracils using the DNA cytidine deaminase apolipoprotein B mRNA editing enzyme catalytic polypeptide-like 3 A (APOBEC3A) (*Carpenter et al., 2012*; *Henry et al., 2009*; *Wijesinghe and Bhagwat, 2012*), and subsequently excised by uracil-specific excision reagent (USER) (*Bitinaite et al., 2007*), resulting in fragmented DNA. Because APOBEC3A also deaminates 5-methyl-2'-deoxycytidine (5mdC) and, to a lesser degree 5-hydroxymethyl-2'-deoxycytidine (5hmdC) (*Sun et al., 2021*), ten-eleven translocation dioxygenase 2 (TET2) and T4 phage β-glucosyltransferase (T4-BGT) can be used to protect 5mdC and 5hmdC prior to APOBEC3A treatment (*Figure 2A*). Both modifications are used as internal control for MetaGPA since the enzymes that catalyzed their formation are well characterized. These enzymes, if present in the sample, are therefore expected to be associated with modifications in our MetaGPA framework.

Effectively, unmodified DNA is depleted and the remaining material in the 'case' cohort should only comprise DNA containing modified cytosines resistant to APOBEC3A treatment. We hypothesize that many forms of cytosine modification, including those unknown to date, are naturally protected from deamination by APOBEC3A, and should be amenable to enrichment by our selection method. Furthermore, the experimental protocol is designed to only protect fully modified DNA, that is DNA for which cytosines are modified irrespective of sequence contexts. A number of such organisms have previously been found, notably in bacteriophages (*Kuo et al., 1968*; *Revel and Georgopoulos, 1969*). We hypothesize that this design would select for DNA-modifying enzymes with little or no sequence specificity.

## Cytosine-modified DNA is retained after enzymatic selection

To demonstrate the feasibility of this approach, a mock community consisting of an equimolar amount of genomic DNA from *Escherichia coli* (containing unmodified cytosine, dC) and T4gT phage (containing 90–95% 5hmdC) was sheared, subjected to enzymatic selection and quantitative PCR (qPCR) was performed to quantify the remaining DNA (see Materials and methods). Compared to the original mock community, enzymatic selection resulted in a 0.5 % recovery of *E. coli* DNA and an average 35 % recovery of T4gT DNA (*Figure 2B*). This result shows that DNA containing modified cytosine is about 70-fold enriched compared to unmodified DNA.

To test the sensitivity and efficiency of this method, we serially diluted modified DNA and unmodified DNA to 1:3, 1:10, 1:100, and 1:1000 molar ratio and carried out the enzymatic selection. Recovery rates were calculated and compared to no-enzyme treatment control. Even at 1:1000 dilution, an average of 48.6 % modified DNA was retained relative to no-enzyme control. This result showed the capability of our method to concentrate trace amounts (picogram-level) of modified DNA from a complex sample (*Figure 2B*).

## MetaGPA association analyses on complex microbiomes identifies candidate enzymes with DNA modification potential

We collected environmental samples for the aim of identifying novel enzymes involved in cytosine modification using MetaGPA. To explore the robustness of MetGPA, we included three environmental samples from distinct sources (*Figure 3A*); of the three samples, two were collected at a sewage treatment plant and the other one was collected from coastal sludge.

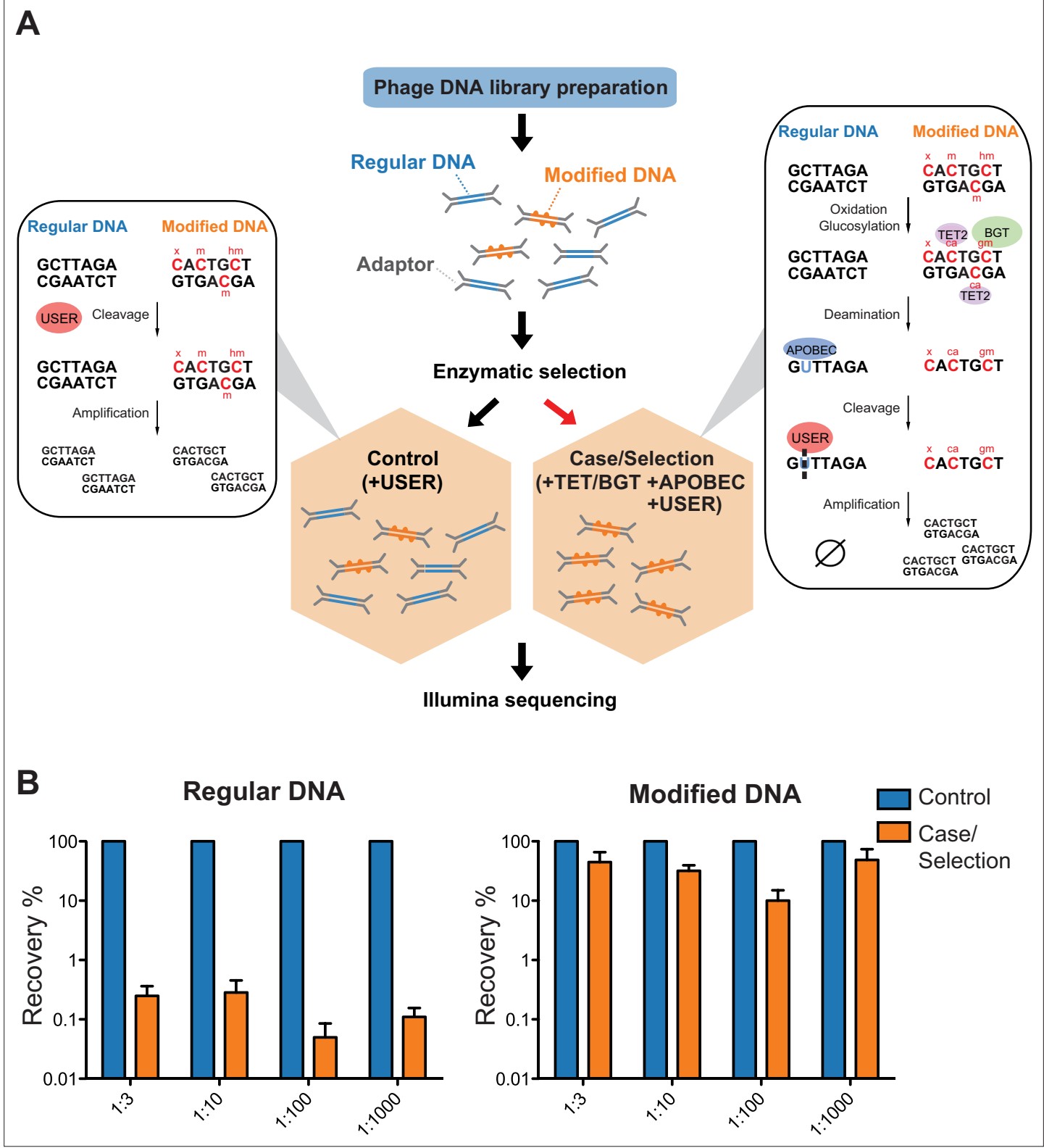

**Figure 2.** Selective sequencing of DNA containing modified cytosines. (**A**), Schematic of enzymatic selection. With enzymatic selection treatment, adaptor ligated products were first incubated with ten-eleven translocation dioxygenase 2 (TET2) and T4 phage β-glucosyltransferase (T4-BGT) so that 5-methyl-2'-deoxycytidine (5mdC) (m) and 5-hydroxymethyl-2'-deoxycytidine (5hmdC) (hm) in these sequences may be oxidized to 5-carboxy-2'-deoxycytidine (ca) or glucosylated to 5-β-glucosyloxymethyl-2'-deoxycytidine (gm) (Materials and methods). Unmodified cytosines were deaminated

*Figure 2 continued on next page*

Figure 2 continued

by APOBEC3A into uracils and then cut by uracil-specific excision reagent (USER), so that DNA with modified cytosines is enriched after the selection. This method is predicted to also preserve unknown forms of cytosine modification (denoted 'x') provided that they block C-to-U deamination. Whereas in the untreated control, only USER is added to clear up damaged DNAs generated in the previous library prep step. (**B**) Enrichment of modified DNA with serial dilutions. A total of 250 ng genomic DNA (from *Escherichia coli*) and cytosine-modified genomic DNA (from T4gt) were used for enzymatic selection or control per assay. Bar graphs show average recovery of DNA from three independent experiments ± SEM.

We aim at obtaining the phage-enriched fraction of each microbiome. The rationale for this selection is based on the fact that phages are known to carry a large diversity of modifications (**Weigele and Raleigh, 2016**) and phages have been shown to fully modify their genomes irrespective of sequencing context, which is a prerequisite for our MetaGPA selection. For this, each sample was precipitated with polyethylene glycol (PEG) followed by DNA extraction using phenol/chloroform (see Materials and methods). Libraries were made in duplicates (except for the coastal sample) using modified adapters to protect them from enzymatic degradation. Libraries were subjected to either enzymatic selection (i.e., case) or no enzymatic selection (i.e., control) prior to amplification (**Figure 3A**). Additionally, spiked-in genomic DNA mixture of *E. coli* (dC), XP12 (5mdC), and T4gt (5hmdC) were added to each sample after adapter ligation and before enzymatic treatment. Recovery of spiked-in modified DNAs was detected as expected (**Figure 3—figure supplement 1A**). Accordingly, we proceeded with Illumina high-throughput sequencing and obtained an average of 60 million paired-end reads per library. Analysis of the read composition reveals consistency of $k$-mer composition between replicates, demonstrating that our enzymatic selection for modified DNA is reproducible (**Figure 3—figure supplement 1B**). Reads from both replicates were combined and normalized $k$-mer frequency plots showed diversity of $k$-mer composition from different sources/samples, while highlighting a small portion of $k$-mers that was highly enriched in the selection libraries (**Figure 3B**).

Next, we apply our newly developed MetaGPA pipeline (**Figure 1—figure supplement 1**) to identify the domain families that are significantly associated with the case cohort. More specifically, reads from the case (selection) and control datasets from the three environmental samples were assembled into contigs. Contigs that were either too short (less than 1000 bp) or redundant were removed (Materials and methods). Reads were mapped to the contigs and the ratio between the normalized coverage in the case (selection) library ($\text{RPKM}_{(case)}$) and the normalized coverage in the control library ($\text{RPKM}_{(control)}$) defines the enrichment score for each contig (Materials and methods, **Figure 3—figure supplement 1C**). A high enrichment score suggests that the contig is derived from DNA containing modified cytosine (i.e., modified contig). We define as modified, contigs with an enrichment score equal or above 3. In total, 3901 modified contigs were identified from three DNA samples (**Figure 3C** and **Figure 3—figure supplement 1C-D**). Distribution of contigs across all length ranges was equal between modified and unmodified contigs (**Figure 3D**) and the proportion of modified contigs among sewage and coastal samples were comparable (about 2 % in both sewage samples and 3.5 % in coastal total contigs) (**Figure 3—figure supplement 1D**).

Annotations of protein domain families were performed using profile hidden Markov models of all protein domain families described in the Pfam database (**El-Gebali et al., 2019**; Materials and methods). Instances of each domain family in the modified and unmodified contigs were calculated and Fisher's exact test was conducted to reveal domain families that were positively associated with DNA modification. Interestingly, there was a high degree of overlap between top associated domain families among different environmental samples (**Figure 4—figure supplement 1A**), suggesting that a group of universal protein families for DNA modification may exist regardless of the origin of the microbiome. Given this consistency, we decided to pool the data from three environmental samples together and repeat the association analysis to achieve higher statistical power. Associated domains were subsequently matched to open reading frames (ORFs) in the modified contigs to define candidate genes (Materials and methods). From this composite dataset, we identified 110 Pfam domain families that were significantly associated with modified contigs (Bonferroni-corrected p-value < 0.01) representing a total of 2365 candidate genes. To estimate the false positive rate of association, we used our two control replicate experiments for which no selection has been done. We then performed a MetGPA analysis with one replicate being the control group and the other replicate being the 'case' group. From the 6737 protein domains assessed in MetaGPA, none of the domains were found significantly associated with the case group (data not shown).

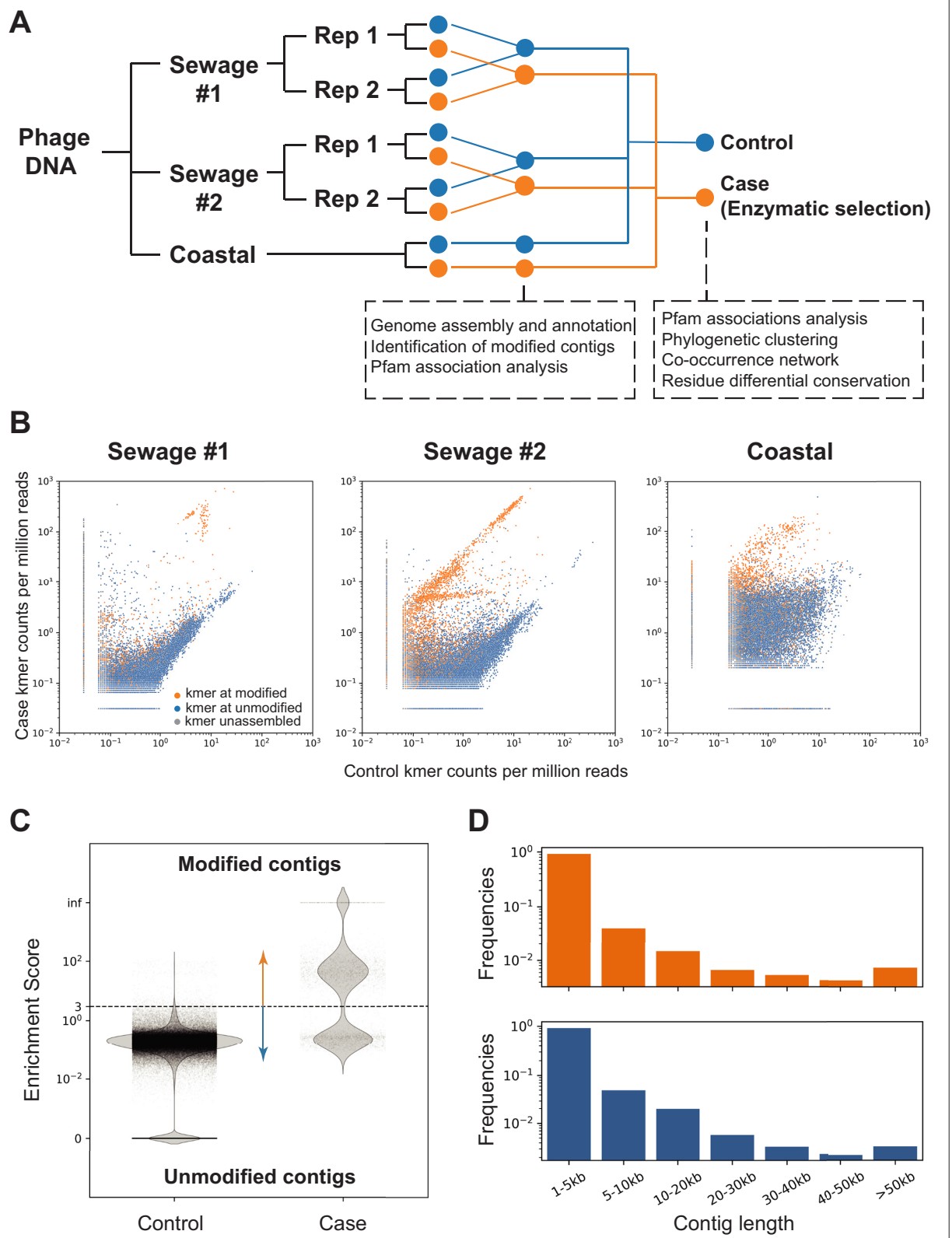

**Figure 3.** Screening of contigs associated with DNA modification. (**A**) Overview of the environmental samples and analysis used in this study. Three independent sequencing experiments were performed on three samples, two sewage samples (same sampling location at different times #1 and #2) and one coastal sample (Beverly, MA). Phage DNA was collected and extracted and for the sewage samples, two technical replicates (Rep 1 and 2) were performed. Analyses performed with separate or composite datasets were indicated in below blocks. (**B**) Normalized frequency of *k*-mer in sequencing

*Figure 3 continued on next page*

*Figure 3 continued*

reads from control (x-axis) compared to case (y-axis) groups. Each dot represents a unique 16-mer and is colored according to the resulting contig category it belonged to. The origin of the k-mer sequences were identified by exact search on the contigs or reverse complement of the contigs. *K*-mers found in the modified contigs were colored in orange; *k*-mers found in unmodified contigs were colored in blue; k-mers not found in contigs were marked as gray. See C below for definition of modified/unmodified contigs. Subsamples of randomly selected 0.1 % of all possible *k*-mers were used for plotting. (**C**) Dotplot represents the enrichment scores for each non-redundant contig in the control (left) and case/enzymatic selection (right) cohorts. Dashed line separates modified from unmodified contigs based on a threshold enrichment score equaled to 3 (enrichment score = RPKM$_{(selection)}$ / RPKM$_{(control)}$). The plot shows every contig from all three environmental samples. (**D**) Distributions of modified (orange) and unmodified (blue) contigs length (in kilobases, kb).

The online version of this article includes the following figure supplement(s) for figure 3:

**Figure supplement 1.** Validation of the selection method for DNA cytosine modification.

The resulting top associations (*Figure 4A* and *Figure 4—figure supplement 1A, B*) contain a number of domain families previously found to be involved in DNA synthesis/modification, validating that this approach can uncover relevant functional domains. For example, a subset of enzymes containing the thymidylate synthase domain (PF00303.20) have been shown to produce hydroxymethylpyrimidines (*Neuhard et al., 1980*). DNA ligase (PF14743.7, PF01068.22), and cytidine/deoxycytidylate deaminase (PF00383.24) are other domains that have been found in DNA-modifying enzymes (*Bhattacharya et al., 1994*; *Subramanya et al., 1996*). In addition, our analysis identified domain families that were not previously known to modify DNA or nucleotides and thus may be novel DNA-modifying enzymes or critical regulators.

To refine the candidate genes, we conducted phylogenetic analysis for each domain family significantly associated with modified contigs. Toward this end, all instances of a particular domain were aligned using a maximum likelihood model and the resulting phylogenetic tree was overlaid with the origin status of the contig (modified/unmodified). Particularly, several domains, including carbamoyltransferase N-terminus (PF02543.16) and C-terminus (PF16861.6) domains, exhibited a clustered pattern in which most of the sequences from modified contigs formed a distinct phylogenetic clade from the other sequences (*Figure 4B* and *Figure 4—figure supplement 1C*). These clades that are almost exclusively derived from modified contigs restating the association of the domain-of-interest with a potential differentiated phenotype of modification. Moreover, this analysis can serve as evidence for refined functional annotation and may suggest a subfamily with specific functions.

Complex DNA modifications in bacteriophages are usually carried out by multiple enzymes whose genes tend to cluster on the genome (*Iyer et al., 2013*). We therefore extended the analysis to study co-occurrences on the same contigs of domain families associated with modification (*Figure 4C* and *Figure 4—figure supplement 2A-C*). We found that, for example, the domains most frequently co-occurring with carbamoyltransferase C-terminus (PF16861.6) were carbamoyltransferase N-terminus (PF02543.16), thymidylate synthase (PF00303.20), phosphoribosyl-ATP pyrophosphohydrolase (PF01503.18), dCMP deaminase Zn-binding region (PF00383.24), and MafB19-like deaminase (PF14437.7) (*Figure 4C*). Congruously, domains belonging to the thymidylate synthase family also co-occurred with carbamoyltransferase N-terminus, phosphoribosyl-ATP pyrophosphohydrolase, dCMP deaminase Zn-binding region, and MafB19-like deaminase domains. Importantly, we found that these co-occurrences are often specific to modified contigs (*Figure 4C* and *Figure 4—figure supplement 2A-C*). For example, the co-occurence of carbamoyltransferase and thymidylate synthase domains is only found in modified contigs (*Figure 4C*). In unmodified contigs, carbamoyltransferase domains were flanked by genes with unrelated functions such as genes encoding for glycosyl transferases group 1 or tRNA N6-adenosine threonylcarbamoyltransferase domains.

Together, these results suggested a multi-domain network related to DNA modification. Unbiased co-occurrence analysis was therefore conducted for the top 20 associated domain families. Indeed, significant positive correlations were observed, and the results demonstrated three interaction cores centered on thymidylate synthases, DNA ligases, and carbamoyltransferases, respectively (*Figure 4D*).

## Differential conservation of amino acids reveals key residues associated with DNA modification

Having demonstrated that MetaGPA can be successfully deployed to identify protein domains associated with a given phenotype, we next considered whether MetaGPA go as far as identifying

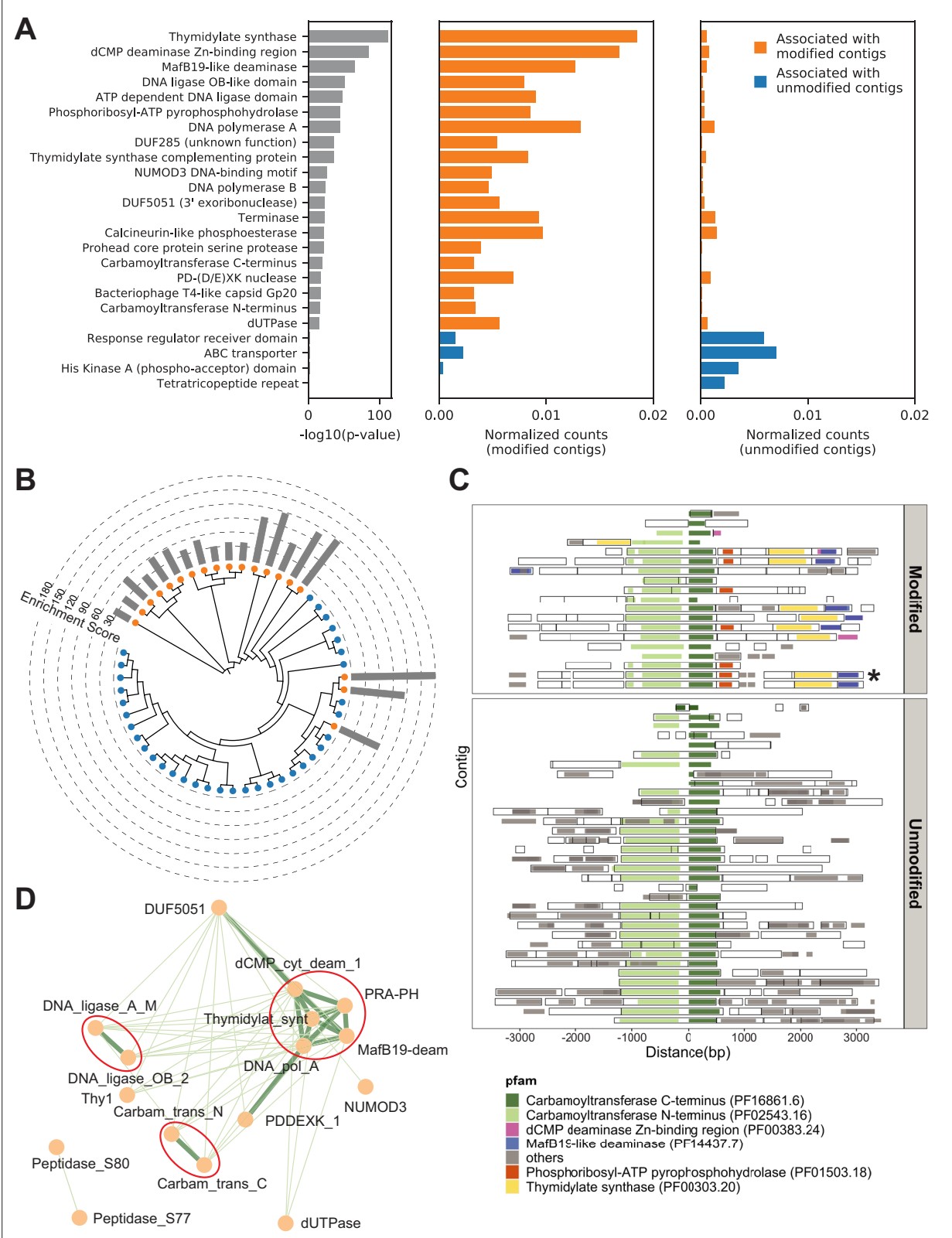

**Figure 4.** Metagenomics Genome-Phenome Association (MetaGPA) study at domain level. (**A**) Identification of associated Pfam protein domains. Left, lists of domains ranked by p-values. Middle and right panels represent the normalized counts of each domain for modified or unmodified contigs, respectively. The top 20 domains significantly associated with modified contigs were colored in orange and the four domains significantly associated with unmodified contigs were colored in blue (Bonferroni-corrected p-value < 0.01). Data were normalized to total counts in each category (sums of

*Figure 4 continued on next page*

*Figure 4 continued*

all domain counts in modified or unmodified contigs), respectively. Domains occurring multiple times on the same contigs were counted only once. (**B**) Phylogenetic tree of carbamoyltransferase C-terminus domain. Orange and blue dots represent modified and unmodified contigs, respectively. Gray bars in the outer ring represent the enrichment scores of each contig with dashed lines showing the scales. Most of the carbamoyltransferase C-terminus domains from modified contigs form a distinctive phylogenetic branch. (**C**) Pfam neighborhood association for carbamoyltransferase C-terminus. Carbamoyltransferase C-terminus is centered in the middle and neighboring Pfams spanning 3 kb upstream and downstream are displayed as solid squares. Colored squares mark the top five Pfams co-occurred with carbamoyltransferase C-terminus in modified contigs. Predicted open reading frames (ORFs) on each contig are marked as hollow squares. Asterisk marks the contig containing the enzyme expressed and purified in *Figure 6*. (D) Correlation networks between the top 20 associated Pfams with modified contigs. Each node represents a Pfam. The thickness and length of edges were based on the p-values between each two nodes with thick and short edges corresponding to more significant relationships (small p-values). Only positive correlations with p-values < 0.05 are shown. The three interaction cores are circled in red.

The online version of this article includes the following figure supplement(s) for figure 4:

**Figure supplement 1.** Metagenomics Genome-Phenome Association (MetaGPA) study reveals significantly associated Pfam domain.

**Figure supplement 2.** Metagenomics Genome-Phenome Association (MetaGPA) study reveals significantly associated Pfam domain.

associations at residue resolution. To identify such associations, we designed a 'differential' conservation score based on existing metrics (*Valdar, 2002*; *Karlin and Brocchieri, 1996*) that reflects the degree of association of individual residue with DNA modification (Materials and methods). This score ranks high the residues that show distinct conservation patterns whether they are derived from modified or unmodified DNA. Conversely, weakly conserved residues or residues conserved invariably between modified and unmodified DNA are ranked low.

We used the thymidylate synthase domain alignments to benchmark the differential conservation score. This well-characterized family of proteins involved in nucleotide biosynthesis is also ranked by MetaGPA as the most significantly associated with DNA modification (*Figure 4A*). The substrate specificity of thymidylate synthases is dictated by the residue at the position 177 (numbering relative to the *E. coli* thymidylate synthase sequence). Previous studies have demonstrated that changing this residue from asparagine (Asn) to aspartate (Asp) can switch the preference of a canonical thymidylate synthase from dUMP to dCMP resulting in the formation of 5mdCMP (*Graves et al., 1992*; *Hardy and Nalivaika, 1992*; *Liu and Santi, 1992*). We therefore hypothesized that position Asn177 should have a high differential conservation score with Asn found conserved in the unmodified contigs and Asp found in modified contigs.

To confirm our hypothesis, we aligned the 433 thymidylate synthase protein sequences identified in our composite dataset with the canonical *E. coli* thymidylate synthase which has been structurally and biochemically characterized (*Stout et al., 1998*). Our differential conservation score identifies position 177 and position 147 (relative to *E. coli* thymidylate synthase) as the top two positions (*Figure 5A*). Both residues are within the enzyme's active site (*Figure 5C*; *Stout et al., 1998*). As expected, residues at positions 177 are mostly Asn for thymidylate synthase protein in the unmodified contigs suggesting that dUMP is the substrate for these enzymes. Conversely, thymidylate synthase proteins in the modified contigs have mostly Asp position 177 consistent with modified cytosines (*Figure 5B*).

The H147 forms a hydrogen bonding network through a water molecule to the keto oxygen of position 4 in dUMP (*Matthews et al., 1990*), suggesting a role in substrate specificity. Alternatively, H147 is part of a H-bond network with Y94, a key residue important for proton abstraction at C5. Therefore, residue occupancy at position 147 may accommodate differences between the pKa of C5 between cytosine and uracil substrates (*Hong et al., 2007*). Taken together, our differential conservation score identified functionally critical residues for thymidylate synthase activity and substrate specificity.

We then performed the same analysis with the top 20 associated Pfam domains from our list (*Figure 4A*). Among them, carbamoyltransferase sequences were aligned with the *O*-carbamoyltransferase family member TobZ that has been structurally and functionally characterized (*Parthier et al., 2012*). Our analysis identified three residues (W408, G421, and R423) with the highest differential conservation scores (*Figure 5—figure supplement 1A-B*). All three residues are located within 14 Å to the carbamoyl phosphate and ADP-binding pocket and may be important in defining the enzyme's specificity toward cytosine derivatives (*Figure 5—figure supplement 1C*).

Altogether, these results indicate that MetaGPA is able to associate a single residue within a protein to the studied phenotype.

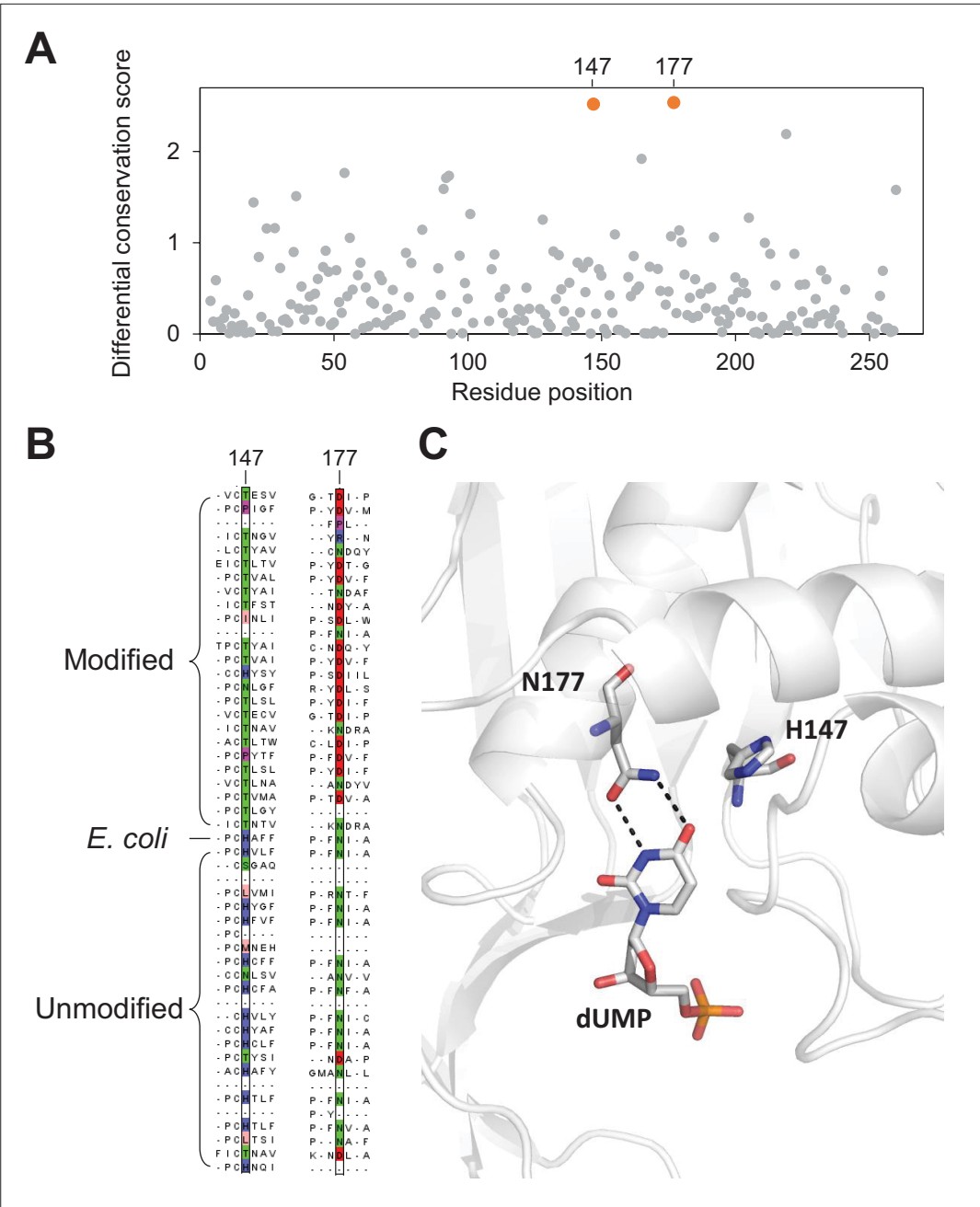

**Figure 5.** Key residues in the thymidylate synthase associated with DNA modification. (**A**) Differential conservation score for each position in the thymidylate synthase alignment. Positions are relative to the *Escherichia coli* thymidylate synthase. (**B**) Multiple sequence alignment of 25 and 24 randomly selected thymidylate synthase sequences from the modified and unmodified contigs respectively together with the *E. coli* thymidylate synthase. Aligned residues at positions 147 and 177 (relative to *E. coli* thymidylate synthase) are colored according to the physicochemical properties of the amino acids. (**C**) Structure of the active site of *E. coli* thymidylate synthase (PDB 1BID) highlighting the H147 and N177 residues and the dUMP substrate.

The online version of this article includes the following figure supplement(s) for figure 5:

**Figure supplement 1.** Association between key residues in the carbamoyltransferase C-terminus domains and DNA modification.

## Characterization of a novel-modifying enzyme – 5-hydroxymethylcytosine carbamoyltransferase

Among the 110 domains that are significantly associated with DNA modifications, carbamoyl-transferase domains are ranking within the top 20 most significant domains in our MetaGPA study (*Figure 4A*) and are exhibiting all three of our refinement metrics. Carbamoyltransferases are part of a large protein family catalyzing the addition of a carbamoyl group to various substrates but so far none of them has been shown to act on any form of cytosine, potentially revealing a new function and a new cytosine modification. We therefore sought to further explore these enzymes for their ability to modify cytosine.

The co-occurrence of carbamoyltransferase and thymidylate synthase homologs specifically in modified contigs (*Figure 4C*) resembles the arrangement of the T4 phages for which genes coding for the dCMP hydroxymethylase and β-glucosyltransferase co-occur on the genome (*Miller et al., 2003*). The T4 dCMP hydroxymethylase is homologous to thymidylate synthase and transfers a carbon from methyltetrahydrofolate (mTHF) to C5 of the pyrimidine ring producing an exocyclic methylene in the active site of the enzyme (*Graves et al., 1992*). However, unlike thymidylate synthase, the methylene intermediate undergoes nucleophilic attack by water producing a hydroxymethyl group. Following incorporation of 5hmC into DNA during replication, T4 β-glucosyltransferase transfers a glucose to the hydroxyl moiety of 5hmC. Thus, the pairing of a carbamoyltransferase with dCMP hydroxymethylase led us to hypothesize a novel form of DNA modification, in which the carbamoyltransferase catalyzes the transfer of a carbamoyl group to the nucleophilic hydroxyl acceptor group of 5hmdC producing 5cmdC (*Figure 6A*).

Our composite dataset contains 62 genes annotated as carbamoyltransferase for which 17 are found in the modified contigs. We selected a carbamoyltransferase candidate gene from a modified contig originally sequenced in sewage #2 sample containing both the thymidylate synthase and the carbamoyltransferase genes (*Figure 6—figure supplement 1A*). The ORF was cloned into pET28b vector, and the 63 kDa protein product was expressed in *E. coli* and purified (*Figure 6—figure supplement 1B*, Materials and Methods). The predicted reaction was tested by enzymatic assays and results showed that every substrate, namely carbamoyl phosphate, ATP, 5hmdC (genomic T4gt DNA was used as substrate in these experiments), and the enzyme were indispensable for the reaction (*Figure 6—figure supplement 1C,D*). The choice of carbamoyl phosphate and ATP substrates were guided by the enzymatic characterization of TobZ previously published (*Parthier et al., 2012*). The expected product was detected by liquid chromatography-mass spectrometry (LC-MS) and confirmed with corresponding fragmentation patterns (*Figure 6—figure supplement 2A, B*). Nearly 70 % of 5hmdC were converted into 5cmdC in the denatured single-stranded T4gt genomic DNA. Interestingly, the activity of our carbamoyltransferase was several fold lower on double-stranded DNA, suggesting the preference of this enzyme for single-stranded DNA (*Figure 6—figure supplement 1C*). When using synthesized single-stranded DNA oligo containing a single internal 5hmdC site as substrate, the conversion rate was nearly 100 % (*Figure 6B*). We also tested if the carbamoyltransferase could react with free deoxynucleoside triphosphate. LC-MS results demonstrated about 60 % conversion of 5-hydroxymethyl-2'-deoxycytidine triphosphate (5hmdCTP) (*Figure 6C*). As expected, no activity was seen for 5-methyl-2'-deoxycytidine triphosphate (5mdCTP) or 5-hydroxymethyl-2'-deoxyuridine triphosphate (5hmdUTP), indicating the carbamoyltransferase is specific to 5hmdCTP. The fact that the carbamoyltransferase is active on 5hmdCTP (*Figure 6C* and *Figure 6—figure supplement 3A*) opens up the possibility that the reaction could take place before the nucleotide is incorporated into the phage DNA.

To examine if the carbamoyltransferase favors certain DNA sequences, we performed the enzymatic assay on a mixed genomic DNA library containing lambda (dC), XP12 (5mdC), and T4gt (5hmdC). Both untreated (control) and treated libraries were subjected to APOBEC3A deamination (see Materials and methods). Carbamoylation protects cytosine derivatives from deamination by APOBEC and thus, the difference in deamination rate between control and treated libraries is indicative of carbamoylation. We saw a decrease in deamination rate only in the T4gt genome indicating specific carbamoylation on 5hmdC. This result further validates the LC-MS results regarding the specificity of the enzyme (*Figure 6D*). Furthermore, all combinations of NCN motifs containing 5hmdC displayed comparable deamination levels compared to the control library, suggesting a general binding mechanism with no preferred context. This result was also consistent with enzymatic assays performed on

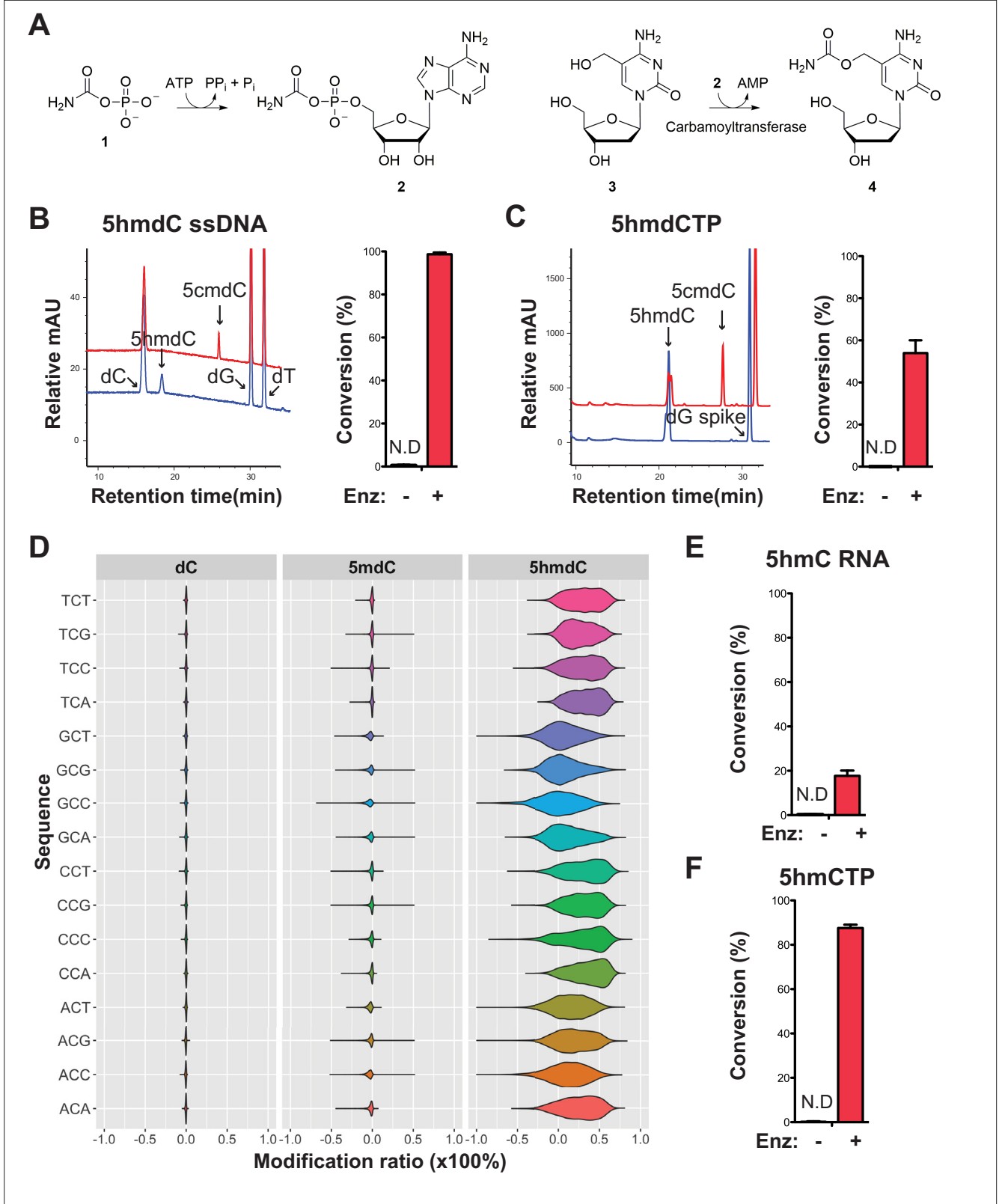

**Figure 6.** Validation of the novel 5-hydroxymethylcytosine (5hmC) carbamoyltransferase. (**A**) Predicted reactions. **1**, carbamoylphosphate; **2**, carbamoyladenylate intermediate; **3**, 5-hydroxymethylcytidine; **4**, 5-carbamoyloxymethylcytidine. (**B**) Liquid chromatography-mass spectrometry (LC-MS) trace and quantification for enzymatic reaction with a single-stranded DNA oligo containing an internal 5-hydroxymethyl-2'-deoxycytidine (5hmdC). Bar graph represents average conversion rates ± SEM from three independent experiments using three different DNA oligos. ND, not detected. (**C**)

*Figure 6 continued on next page*

*Figure 6 continued*

LC-MS trace and quantification for enzymatic reaction with 5-hydroxymethyl-2'-deoxycytidine triphosphate (5hmdCTP) . Bar graph shows average conversion rates ± SEM from three independent experiments. (**D**) Enzyme preference on NCN sequences. Genomic DNA from lambda (dC), XP12 (5mdC), and T4gt (5hmdC) were mixed at 1:1:1 ratio by molarity before being subjected to enzymatic selection. Modified ratio of each C in NCN motifs (with N being A, T, C, or G) was normalized to the no-enzyme control. (**E**) LC-MS quantification for enzymatic reaction with a single-stranded RNA oligo containing an internal 5hmC. Bar graph shows average conversion rates ± SEM from three independent experiments. (**F**) Quantification for enzymatic reaction with 5-hydroxymethylcytidine triphosphate (5hmCTP). Bar graph shows average conversion rates ± SEM from three independent experiments. Oligonucleotides and nucleoside triphosphate were converted to nucleosides before LC-MS analysis. A dG spike was added as internal reference for quantification of nucleoside triphosphates.

The online version of this article includes the following figure supplement(s) for figure 6:

**Figure supplement 1.** Validation of the novel 5-hydroxymethylcytosine carbamoyltransferase.

**Figure supplement 2.** Validation of the novel 5-hydroxymethylcytosine carbamoyltransferase.

**Figure supplement 3.** Validation of the novel 5-hydroxymethylcytosine carbamoyltransferase.

free nucleotides, which further suggests that the in vivo carbamoylation reaction may take place prior to DNA replication.

Based on our finding that the carbamoyltransferase prefers single-stranded DNA, we further investigated whether RNA containing 5-hydroxymethylcytosine can be modified by the enzyme. LC-MS and fragmentation pattern confirmed that 5cmC is formed during the reaction, albeit at much lower yields (*Figure 6E* and *Figure 6—figure supplement 3B-C*). Carbamoylation of free nucleotide 5-hydroxymethylcytidine triphosphate (5hmCTP) was also detected (*Figure 6F* and *Figure 6—figure supplement 3A*). We thus concluded that this novel nucleic acid-modifying enzyme identified from MetaGPA studies acts on DNA, RNA, and nucleotide triphosphates.

## Discussion

In this work, we report a novel analytic framework called MetaGPA that links functional phenotype with genetic information in metagenomes. Being cost effective and requiring reduced efforts, MetaGPA essentially rests on two sequencing libraries derived from the same starting material: one library reflects all the organisms in the community while the other library results from a selection to only contain organisms with a phenotype of interest. This experimental setup allows for candidate genes identification in less than a week.

Our MetaGPA study framework is phenotype-driven and the identification of candidate genes is agnostic to prior annotations. Theoretically, it can be performed with any case/control cohort pair as long as distinct phenotypes can be partitioned through selection of the case cohort. For example, these phenotypes may include but are not limited to DNA modifications, phage sensitivity to chemicals, and cell surface determinants such as O-antigen. Nonetheless, this partitioning is crucial for MetaGPA to succeed and therefore requires the development and optimization of the selection process for every new MetGPA application.

We used this approach to screen environmental phage metagenomes and successfully identified 110 domains representing 2365 candidate proteins, mostly enzymes that are significantly associated with DNA modifications. From these candidates, we validated a novel DNA/RNA-modifying enzyme responsible for the previously unknown 5cmdC/5cmC modification. To our knowledge, the only reported carbamoylated nucleotide is the 5-carbamoylmethyluridine, which is located in the first position of the anticodon of yeast valine tRNA34 (*Yamamoto et al., 1985*). Importantly, we predicted, at single residue resolution, the key positions in the enzyme associated with this DNA modification, enabling candidate prioritization and protein engineering.

Given the co-occurrence on the same contig of phage-specific genes (such as phage tail genes) with the 5-hydroxymethylcytosine carbamoyltransferase gene characterized in this study, the 5cmdC modification is most likely found in a bacteriophage. Because of the nature of the MetaGPA selection and the fact that the carbamoyltransferase show no sequence specificity, we predict that all the cytosines in this phage are modified, possibly conferring the ability to escape most of the restriction digestion system found in bacteria. This evolutionary advantage may explain the prevalence of this enzyme family in the phage fraction of all the microbiomes investigated in this study.

In these MetaGPA experiments, the sequencing libraries can be directly used as material for selection because the phenotype of interest (cytosine modification) is covalently attached to the genetic material. As such, scenarios for which the phenotype is covalently attached to the genetic material are the most straightforward applications of MetaGPA. However, for other scenarios for which phenotypes are not physically coupled with DNA or RNA, selection will have to be done while preserving the integrity of cells and viral particles to retain the link between phenotype and genotype. Thus, the limited availability of adequate selection processes that preserve such links may restrict the broader applicability of MetaGPA. While some selections can easily be adapted to MetaGPA, others may turn out to be difficult to implement.

For those applications for which a selection can be implemented, MetaGPA provides notable benefit for de novo exploratory discoveries especially with largely unknown microbial communities such as the virulome where existing knowledge is limited.

## Data access
All raw and processed sequencing data generated in this study have been submitted to the NCBI Sequence Read Archive (SRA; https://www.ncbi.nlm.nih.gov/sra) under accession number PRJNA714147.

# Materials and methods
## Genomic DNA
The *E. coli* K-12 MG1655, XP12, and T4gt genomic DNA used in this study were obtained from NEB.

## Environmental phage collection
For each batch, 2–4 L of sewage or coastal seawater were used for phage collection. Large debris and bacterial cells were pelleted and removed by centrifuging at 5000× *g* for 30 min at 4 °C. Phage particles in the supernatant were precipitated by adding PEG8000 to 10% (w/v) and NaCl to 1 M and let stand at 4 °C overnight. Aggregates of phage particles were pelleted at 10,000× *g* for 30 min at 4 °C, washed with 1 mL solution containing 10 % PEG8000 and 1 M NaCl, and resuspended in 2–4 mL of phage dilution buffer (10 mM Tris-HCl at pH 8.0, 10 mM $MgCl_2$, 75 mM NaCl). The crude phage particle suspension was stored at 4 °C for subsequent phenol-chloroform DNA extraction.

## Phenol-chloroform DNA extraction
Two to four milliliters of crude phage suspension was divided in 400 µL aliquots. For each aliquot, phage particles were lysed at 56 °C for 2 hr in 550 µL of lysis buffer (100 mM Tris-HCl at pH 8.0, 27.3 mM EDTA, 2 % SDS, ~1.6 U Proteinase K [NEB #P8107]). After lysis, RNase A was added to 10 µg/mL and the reaction was incubated at 37 °C for 30 min; 1 × volume (~550 µL) of phenol-chloroform (Tris-HCl buffered at pH 8.0) was mixed with the lysis solution and vortexed vigorously for ~1 min, and centrifuged at 10,000× *g* for 5 min for phase separation. The top aqueous layer (~500 µL) was collected and mixed with 1 × volume (~500 µL) of chloroform, vortex vigorously, and centrifuged for phase separation. The top aqueous layer (~450 µL) was collected; 1 × volume of isopropanol was slowly added on top of the aqueous solution. Phage DNA was 'spooled' with a glass capillary by swirling and mixing isopropanol with the aqueous solution. The spooled DNA was washed in 70 % ethanol, dried at room temperature for ~30 min, and dissolved in ~600–800 µL of TE buffer (10 mM Tris pH 7.5, 1 mM EDTA).

The phage DNA solution was further purified by ethanol precipitation. Briefly, DNA was precipitated by adding 0.1 × volume of 3 M sodium acetate and 2.5 × volume of 100 % ethanol and incubated at −20 °C overnight. Precipitated DNA was pelleted at 16,000× *g* for 20 min, washed twice with 1 mL of 70 % ethanol, dried at room temperature, and finally dissolved in 200 µL of TE buffer for storage at −20 °C. On average more than 20 µg of DNA was extracted in each batch.

## Illumina library preparation
For each environmental sample, 1 µg of metagenomic DNA was sheared to 300 bp in 130 µL of TE buffer (10 mM Tris pH 7.5, 1 mM EDTA) using Covaris S2 Focused Ultrasonicator; 1.3 µL of 10 mg/mL RNase A (Qiagen #1007885) was added and incubated at 37 °C for 30 min to remove RNA. To remove

EDTA, the sheared DNA was purified with Zymo Oligo Clean & Concentrator Kit (#D4061) and eluted in 50 µL of 1 mM Tris buffer (pH 7.5).

One reaction of NEBNext Ultra II DNA Library Prep Kit for Illumina (NEB #E7645) was used for 1 µg of input DNA, with the following modifications to the standard protocol: 5mdC Y-shaped Illumina adaptors were used to protect the adaptor from subsequent enzymatic treatment. The dCTP was replaced with 5mdCTP in the end repair reaction (5mdCTP was used instead of regular dCTP to protect end-repaired fragments from subsequent enzymatic treatment).

The DNA library was purified with 1 × volume of NEBNext Sample Purification Beads (NEB #E7103) and eluted with 40 µL of 1 mM Tris buffer (pH 7.5).

For each of the two sewage DNA samples, experiments were performed in duplicate. Each one contained two pairs of replicate libraries subjected to enzymatic selection or control, respectively. The coastal sample generated only one pair: one library for enzymatic selection and one for control.

## Enzymatic selection protocol

First to test the recovery of modified DNA with enzymatic selection, mixed genomic DNA (*E. coli* and T4gt) were prepared at various dilutions. A total of 250 ng mixed DNA was used per reaction; 1 µL TET2 (NEB #7120 S) and 1 µL T4-BGT (NEB #M0357S) were added to the 50 µL reaction mixture containing 1 × TET2 reaction buffer, 40 µM UDP-glucose, and 40 µM iron(II) sulfate hexahydrate. After 60 min incubation at 37 °C, proteinase K was added at 0.4 mg/mL to inactivate the enzymes. Products were purified with Zymo Oligo Clean & Concentrator kit (#D4061) and eluted in 16 µL water. To denature double-stranded DNA, 4 µL formamide (Sigma #11814320001) was added. The 20 µL mixture was then incubated at 95 °C for 10 min and immediately transferred to an ice bath. One µL APOBEC3A (NEB #E7120S) was added directly to the reaction with 10 µL of 10 × APOBEC3A reaction buffer and 1 µL BSA (10 mg/mL). The reaction volume was brought up to 100 µL with water. APOBEC3A-mediated deamination was conducted at 37 °C for 3 hr. Purification was performed using Zymo Oligo Clean & Concentrator kit and elution with 43 µL of water. In the final step, the reaction was incubated with 2 µL of USER (NEB #5508) in 1 × CutSmart Buffer at 37 °C for 15 min before final purification with Zymo Oligo Clean & Concentrator kit. Purified samples were then used for qPCR in the next step.

For each prepared phage library sample, 100 ng spiked-in genomic DNA mixture (*E. coli*:XP12:T4gt = 1:1:1 by molarity) were added to the library before being subjected to enzymatic selection protocol described above. The final library was eluted in 50 µL of 1 mM Tris buffer (pH 7.5).

## Quantitative PCR

The qPCR reactions were performed with enzymatic selection or control samples using Luna Universal qPCR Master Mix (NEB #M3003S) on a Bio-Rad CFX96 real-time PCR detection system. Two µL of purified DNA (diluted 100-folds) were added per reaction. Primers used in the experiments were the following: *E. coli* F: 5'-TTGCTGAGTTTCACGCTTGC, *E. coli* R: 5'-AAAACCGCTTGTGGATTGCC, T4gt F: 5'-TCGCGAAACGGTTTTCCAAG, T4gt R: 5'-AAAGCGCTTGACCCAACAAC, XP12 F: 5'-TGCGAT-GTTGGATTCGTTGG, and XP12 R: 5'-ACAACCCGCCATAATGGAAC. Recovery was normalized to control using the delta-delta Ct method.

## Illumina sequencing

Libraries were indexed (with NEBNext Multiplex Oligos for Illumina E7335), amplified using NEBNext Ultra II Q5 Master Mix (six cycles for control library and 12 cycles for selection library) and pooled for sequencing on an Illumina Nextseq instrument with paired end reads of 75 bp.

### Reads trimming and *k*-mer analysis

Paired-end reads were downloaded as FASTQ files and trimmed with Trim Galore v0.6.4 (https://www.bioinformatics.babraham.ac.uk/projects/trim_galore/) using the `--paired` option. *k*-mer counting from reads was done with JELLYFISH v2.2.10 (*Marçais and Kingsford, 2011*) and 16-mer was chosen based on best resolution.

## Computational workflow

The complete workflow contains the core MetaPGA pipeline and additional association analyses. The main steps included in the core pipeline are the following: first, trimmed reads are processed and mapped to contigs generated by de novo assembly. Annotation of protein domains/Pfams on each assembled contigs was also done at this step. Second, using the normalized mapped reads, the enrichment scores for every assembled contigs were calculated and modified contigs were selected based on the chosen threshold. Then, we applied statistical analysis to every protein domain/Pfam to determine if it is significantly associated with the modification phenotype. Hence, a list of candidate protein domains associated with the modification are generated. Last, MetaGPA provides three different types of association analysis (phylogenetic clustering analysis, residue differential conservation analysis, and multi-domain co-occurrence network analysis) to further guide if a custom chosen protein domain is a good candidate. Detailed methodology for each step is provided below. The scripts and detailed documentation of the pipeline are available at https://github.com/linyc74/MetaGPA, (*Lin, 2021*, copy archived at swh:1:rev:28c23f47fcf108b0e7b8851b92f37921358c8e8e).

## Sequencing data processing

De novo assembly of contigs for each sample was performed with SPAdes v3.13.0 (*Nurk et al., 2017*) with the `--meta option`. We selectively reported contigs longer or equal to 1000 bp. To remove redundant contigs between case and control samples, we used CD-HIT v4.8.1 (*Fu et al., 2012*) nucleotide mode cd-hit-est with sequence identity threshold set to 0.95 (sequences with more than 95 % similarity are considered redundant). We set this threshold due to the fact that many microorganism genomes are related. Other options used were -n 10 -d 0 M 0 T 4. The remaining non-redundant contigs were annotated with HMM-based Pfam entries (Pfam-A) using HMMER v3.3 (http://hmmer.org/). Alignment of reads onto contigs was done with BOWTIE2 v2.3.5.1 (*Langmead and Salzberg, 2012*) together with SAMTOOLS v1.9 (*Li et al., 2009*) to generate, sort, and index bam files for later analysis. ORFs on each contig were predicted with GLIMMER 3.02 (*Delcher et al., 2007*). We used the long-ORFs program with options `--cutoff` 1.15 and `--linear` to identify long ORFs that are very likely to contain genes. Other options used in glimmer3 program were `--max_olap 100`, `--gene_len 110`, and `--threshold` 30.

## Contig enrichment score calculation

The enrichment score for each contig was calculated using the normalized mapped reads (reads per kb per million, RPKM) from selection and control as follows: enrichment score = $RPKM_{(selection)}$ / $RPKM_{(control)}$. The mapped reads counts were generated with Multicov using BEDTOOLS v2.29.2 (*Quinlan and Hall, 2010*). Contigs with higher enrichment score represent more mapped reads in case library relative to control library, therefore, are more likely to be associated with modification. We considered contigs with an enrichment score greater or equal to three to be modified and the rest unmodified. The calculation was done individually for three independent experiments.

## Fisher's exact test and correction to determine associated Pfams

The information including the number and type of Pfams on each contig was obtained with hmmsearch in the annotation step. We then separately counted the numbers of modified and unmodified contigs containing each type of Pfam. To avoid redundant counting, Pfams occurred multiple times on the same contig were counted only once. Fisher's exact test was performed for each Pfam to identify if the count difference between the modified and unmodified contig group is significant. Because large-scale multiple testing was conducted for each Pfam, we did the Bonferroni correction to adjust the p-value. Both tests were performed in python with SciPy or Statsmodels modules.

## Phylogenetic analysis

For each Pfam of interest, the protein sequences from contigs containing the Pfam were aligned with MUSCLE v3.8.1551 (*Edgar, 2004*). The resulting aligned fasta files were subjected to construct phylogenetic trees using the maximum likelihood method in the phylogenetic analysis program RAxML v8.2.12 (*Stamatakis, 2014*). We chose the -f a option to do rapid bootstrap analysis and the -m PROTGAMMAAUTO model to automatically determine the best protein substitution model to be used for

the dataset. The parsimony trees were built with random seeds 1237. The online tool iTOL (https://itol.embl.de/) was used to visualize trees.

## Co-occurrence network analysis

The presence-absence matrix with rows being the Pfams and columns being the contigs was generated with annotation output file from the previous step. We specifically performed co-occurrence analysis in the R package co-ocur v1.3 (*Griffith et al., 2016*) for the top 20 Pfams associated with modified contigs. Significant positive correlations (p-value < 0.05) were exported and the network was visualized in Cytoscape v3.8.0 (*Shannon et al., 2003*) with prefuse force directed layout.

## Differential conservation score

Protein sequences were assigned to two groups according to whether they were encoded on modified or unmodified DNA. After multiple sequence alignment, positions that have less than 50 % residues present were ignored. Differential conservation score was calculated at each aligned position. For each position in the alignment, intra-group similarity scores were calculated by the average of all possible 'within-group' pairwise similarities, while the inter-group similarity score was calculated from all possible 'across-group' pairwise similarities using the BLOSUM80 matrix. For a given multiple sequence alignment column, let $N_1$ and $N_2$ be the number of residues for the modified and unmodified groups, respectively, the two intra-group similarity scores ($I_{modified}$ and $I_{unmodified}$) were defined as:

$$I_{modified} = \sum_{i=1}^{N_1} \sum_{j>i}^{N_1} M\left(a_i, a_j\right) \times \frac{2}{N_1\left(N_1-1\right)}$$

$$I_{unmodified} = \sum_{i=1}^{N_2} \sum_{j>i}^{N_2} M\left(a_i, a_j\right) \times \frac{2}{N_2\left(N_2-1\right)}$$

where $M\left(a_i, a_j\right)$ is the value of amino acid pair $a_i$ and $a_j$ in the BLOSUM80 matrix. The inter-group similarity score ($J$) was defined as:

$$J = \sum_{i=1}^{N_1} \sum_{j=1}^{N_2} M\left(a_i, a_j\right) \times \frac{1}{N_1 N_2}$$

The differential conservation score ($S$) was defined as the average of two intra-group similarity scores subtracted by the inter-group similarity score.

$$S = \frac{I_{modified} + I_{unmodified}}{2} - J$$

## Expression and purification of carbamoyltransferase

The following carbamoyltransferase sequence : MSDLLLTLGHNASAIAISVGDDGAAKVENAYELERL TGKKSDSAFPIDAIIALKERGMDKIDRVYVSHWSPTGRVDDLKAKYWDRSIFPPHVPVITQESMNLTHHDCH AQAAMAFAGSSFPTKDTGVLVVDGFGNLAEHLSYYRVQAGGQLHLMRRWYGYGTSLGLMYQYATSFLG LKMHEDEYKLLGYGARVATIGCDMDVLNQRIFTEAQAFLKRFRSLNSFEMSPDLAGLPAVQEKWAERF AAILDDVGFKGSSSTYEARCIVGYAVQQLLEIVIRNLFMADLPKPTNLIVTGGVAFNVELNRMLLGLIPGKL CVMPLAGDQGNALGLWAFSNRRAKLDFGDLCWGRREMTLGEPGPDTPLPDGMIVVEHDTPAVYEAIAE QLKTVGFINIVRGNMEFGPRALCNTTTLARADDRAVVEEINRINGRDTVMPFAPVVSAHEWLRYFPDA SRLHRSAEFMICAVQYAPGLGEQVPGAALRTVKGLYTGRPQVYSSKYEWDSVTRILDDYGLLINTSFNVHGV PICLDLKHVVHSHQFQRERNPNVRTIVIAN* was extracted from de novo assembled contigs. The expression plasmid was synthesized from GenScript. Two 6 × His-tags were co-expressed at both the N-terminus and the C-terminus of the recombinant protein using T7 Express Competent *E. coli* (NEB #C2566). Cells were cultured in LB media until an OD600 of 0.6 was reached and induced with 0.4 mM IPTG (Growcells #MESP-2002) for protein expression. One µM iron(II) was also added to facilitate protein folding. The induced cultures were maintained at 16 °C in a shaker at 200 rpm for 23 hr. Cells were harvested by spinning down cell pellets at 3500 rpm at 4 °C for 30 min. Cell pellets from 4 L culture were resuspended in 160 mL buffer A containing 20 mM Tris pH 7.5, 500 mM NaCl, 0.05 % Tween-20, 20 mM imidazole, and sonicated using a Misonix S-4000 sonicator with 20 s on and 20 s off cycles until an OD260 plateau was reached. Cell lysates were spinned down at 13,000 rpm for 30 min in a pre-chilled centrifuge at 4 °C. The supernatant was separated and combined with 0.2 mM PMSF (Sigma #78830); 50 mL of supernatant was loaded on AKTA (GE Healthcare Life Sciences) with 1 mL

Histrap column (GE Healthcare Life Sciences) pre-equilibrated with buffer A. The column was washed with 50 mL buffer A and eluted with a gradient of buffer B containing 20 mM Tris pH 7.5, 500 mM NaCl, 0.05 % Tween-20, and 500 mM imidazole. Aliquots containing concentrated proteins were pooled and diluted 1:1 with 20 mM Tris pH 7.5, 5 % glycerol and 0.05 % Tween-20. The diluent was reloaded on AKTA with 5 mL Hitrap Q HP column (GE Healthcare Life Sciences), followed by a wash with 35 mL buffer containing 20 mM Tris pH 7.5, 100 mM NaCl, 5 % glycerol, and 0.05 % Tween-20 and eluted with a gradient of buffer containing 20 mM Tris pH 7.5, 1 M NaCl, 5 % glycerol, and 0.05 % Tween-20. Finally, collected fractions with concentrated proteins were pooled and mixed with equal volume glycerol for storage at −20 °C.

## Carbamoyltransferase enzyme assay

For enzyme assay using T4gt genomic DNA as substrate, 10 min incubation at 95 °C was performed to denature double-stranded DNA and the sample was immediately transferred to ice bath to prevent re-annealing. Then 0.38 nM denatured DNA was used for each 50 μL reaction with 1 × NEBBuffer2.1 (NEB #B7202S), freshly prepared 10 μM iron(II) sulfate hexahydrate (Sigma #203505), freshly prepared 10 mM carbamoylphosphate and 5 mM ATP. Carbamoyltransferase was added to the reaction at 7.2 μM. The reaction mixture was incubated at 30 °C for 3 hr before adding 2 μL Proteinase K to inactivate the enzyme. After 30 min incubation at 37 °C with proteinase K, DNA was purified with Zymo Oligo Clean & Concentrator kit. For assays with synthesized single-stranded DNA oligos containing 5hmdC, the heat-denaturing step was omitted. Oligos were added at 1.6 μM per 50 μL reaction with the same concentration of carbamoyltransferase and other components added as listed before. Purification was performed using Norgenbiotek Oligo Clean-up and Concentrator kit (#34100). For assays with free nucleotides, 0.5 mM of the corresponding nucleotide was used per reaction. For assays with synthesized RNA oligos containing 5hmC, 1.57 μM RNA was added per reaction.

## LC-MS and fragmentation analysis

Genomic DNA and synthetic oligonucleotides were digested to nucleosides by treatment with the Nucleoside Digestion Mix (NEB #M0649S) at 37 °C for 3 hr. The resulting nucleoside mixtures were directly analyzed by reversed-phase LC/MS or LC-MS/MS without further purification. Nucleoside and nucleotide analyses were performed on an Agilent LC/MS System 1200 Series instrument equipped with a G1315D diode array detector and a 6120 Single Quadrupole Mass Detector operating in positive (+ESI) and negative (-ESI) electrospray ionization modes. LC was carried out on a Waters Atlantis T3 column (4.6 mm × 150 mm, 3 μm) at a flow rate of 0.5 mL/min with a gradient mobile phase consisting of 10 mM aqueous ammonium acetate (pH 4.5) and methanol. MS data acquisition was recorded in total ion chromatogram mode. LC-MS/MS was performed on an Agilent 1290 UHPLC equipped with a G4212A diode array detector and a 6490 A triple quadrupole mass detector operating in the positive electrospray ionization mode (+ESI). UHPLC was performed on a Waters XSelect HSS T3 XP column (2.1 × 100 mm, 2.5 μm particle size) at a flow rate of 0.6 mL/min with a binary with a gradient mobile phase consisting of 10 mM aqueous ammonium formate (pH 4.4) and methanol. MS/MS fragmentation spectra were obtained by collision-induced dissociation in the positive product ion mode with the following parameters: gas temperature 230 °C, gas flow 13 L/min, nebulizer 40 psi, sheath gas temperature 400 °C, sheath gas flow 12 L/min, capillary voltage 3 kV, nozzle voltage 0 kV, and collision energy 5–65 V.

## Sequence preference of carbamoyltransferase

Library preparation was performed except the following modifications to the standard protocol: (1) we did not perform RNase A treatment for this experiment; (2) we used pyrrolo-dC Y-shaped adaptor instead of regular adaptor so that they are protected from subsequent enzymatic treatment. For each library, 1 μg genomic DNA mixture (Lambda:XP12:T4gt = 1:1:1 by molarity) was used. After adapter ligation, DNA libraries were purified with 1 × volume of NEBNext Sample Purification Beads (NEB #E7103) and eluted with 20 μL nuclease-free water. The sample was then denatured by heating to 95 °C for 10 min and subjected to carbamoyltransferase reaction as described above. Carbamoyltransferase was added to the reaction at 7.2 μM for every 1 μg DNA library. Purification was performed using Zymo Oligo Clean & Concentrator kit and eluted with 16 μL water. Purified DNA samples were heated at 90 °C with 4 μL formamide to generate single-stranded fragments for the deamination

reaction. One µL APOBEC3A was added per reaction to both carbamoyltransferase-treated or control (untreated) samples with 10 µL of 10 × APOBEC3A reaction buffer and 1 µL BSA (10 mg/mL). The reaction mixture was incubated at 37 °C overnight. Final libraries were purified using Zymo Clean & Concentrator kit, indexed (with NEBNext Multiplex Oligos for Illumina E7335) and amplified with NEBNext Q5U Mater mix (NEB #M0597). Sequencing was performed on an Illumina Mi-seq instrument with pair-end reads (2 × 75 bp). Raw reads were trimmed with TrimGalore. Methylation was analyzed with Bismark v0.22.3 and plotted with RStudio v3.6.3.

## Synthesis of 5hmC RNA oligonucleotide

Forward and reverse DNA templates were annealed at 95 °C for 4 min and slowly cooled for 20 min. RNA synthesis was performed with HiScribe T7 High Yield RNA Synthesis Kit (NEB #E2040). One µg of annealed DNA template was used per reaction with 1.5 µL T7 RNA Polymerase Mix. 5hmCTP was used with the other three nucleotides ATP, UTP, and GTP at 7.5 mM each. The reaction was incubated at 37 °C for 4 hr. Two µL nulease-free DNase I were added to each reaction to digest DNA templates, followed by incubation at 37 °C for 15 min. Synthesized RNA was purified with Norgenbiotek Oligo and Concentrator kit and stored at −80 °C.

## Nucleotides and synthesized oligos

Single-stranded DNA oligos used in enzymatic assays were purchased from IDT. The sequences are as follows:

> 5hmdC-1: 5'-TGTCCGATAGACT{5hmdC}TACGCA;
> 5hmdC-2: 5'-AACTCGCCGAGGATTT{5hmdC}TAC;
> 5hmdC-3: 5'-{Fam-AmC6}ACACCCATCACATTTACAC{5hmdC}GGGAAAGAGTTGAATGTAGAGTTGG.

The DNA templates for synthesizing RNA were purchased from IDT as follows (T7 promoter sequence was underlined):

> Forward: 5'-GACCTAATACGACTCACTATAGGGAGTGAGAAGATGGTCTAGGTGTTTATTGGTGATGAA.
> ComRev: 5'-TTCATCACCAATAAACACCTAGACCATCTTCTCACTCCCTATAGTGAGTCGTATTAGGTC.

5hmdCTP (D1045) and 5mdCTP (D1035) were purchased from Zymo Research. 5hmdUTP (N-2059) and 5hmCTP (N-1087) were purchased from Trilink Biotechnologies.

## Code availability

Custom-built bioinformatics pipelines are available at https://github.com/linyc74/MetaGPA (*Lin, 2021*, copy archived at swh:1:rev:28c23f47fcf108b0e7b8851b92f37921358c8e8e).

## Acknowledgements

The authors would like to thank Barry Cohen and the junior team (Grace, Alina, Camille, and Jeanne) for environment sample collections, Lana Saleh for useful discussions and providing the RNA oligos, Vladimir Potapov, Zhiyi Sun, and Tamas Vincze for analysis and IT support and the sequencing core. John Buswell for providing the Pyrrolo dC adaptor. This work was supported by New England Biolabs.

## Additional information

### Competing interests

Weiwei Yang, Nan Dai, Romualdas Vaisvila, Peter Weigele, Yan-Jiun Lee, Ivan R Corrêa, Ira Schildkraut, Laurence Ettwiller: is an employee of New England Biolabs Inc, a manufacturer of restriction enzymes and molecular reagents. Yu-Cheng Lin, William Johnson: was an employee of New England Biolabs Inc, a manufacturer of restriction enzymes and molecular reagents.

## Funding

| Funder | Grant reference number | Author |
|---|---|---|
| New England Biolabs | no data | Weiwei Yang<br>Yu-Cheng Lin<br>William Johnson<br>Nan Dai<br>Romualdas Vaisvila<br>Peter Weigele<br>Yan-Jiun Lee<br>Ivan R Corrêa<br>Ira Schildkraut<br>Laurence Ettwiller |

The funders had no role in study design, data collection and interpretation, or the decision to submit the work for publication.

## Author contributions

Weiwei Yang, Data curation, Formal analysis, Investigation, Methodology, Project administration, Software, Validation, Visualization, Writing – original draft, Writing – review and editing; Yu-Cheng Lin, Conceptualization, Data curation, Formal analysis, Investigation, Methodology, Resources, Software, Supervision, Validation, Visualization, Writing – original draft, Writing – review and editing; William Johnson, Investigation, Methodology; Nan Dai, Investigation, Methodology, Validation, Writing – review and editing; Romualdas Vaisvila, Conceptualization, Investigation, Methodology, Writing – review and editing; Peter Weigele, Conceptualization, Supervision, Writing – review and editing; Yan-Jiun Lee, Formal analysis, Investigation, Methodology, Writing – review and editing; Ivan R Corrêa, Conceptualization, Data curation, Supervision, Writing – review and editing; Ira Schildkraut, Investigation, Methodology, Validation; Laurence Ettwiller, Conceptualization, Data curation, Formal analysis, Investigation, Methodology, Project administration, Supervision, Validation, Writing – original draft, Writing – review and editing

## Author ORCIDs

Weiwei Yang http://orcid.org/0000-0002-8836-9018
Yu-Cheng Lin http://orcid.org/0000-0002-4787-2565
Ivan R Corrêa Jr, http://orcid.org/0000-0002-3169-6878
Laurence Ettwiller http://orcid.org/0000-0002-3957-6539

## Decision letter and Author response

Decision letter https://doi.org/10.7554/eLife.70021.sa1
Author response https://doi.org/10.7554/eLife.70021.sa2

# Additional files

## Supplementary files

- Supplementary file 1. Amounts of DNA used for enzymatic selection sensitivity test.
- Supplementary file 2. Pfam associations with DNA modification.
- Transparent reporting form
- Source data 1. Supplemental source data.

## Data availability

All raw and processed sequencing data generated in this study have been submitted to the NCBI Sequence Read Archive (SRA; https://www.ncbi.nlm.nih.gov/sra) under accession number PRJNA714147.

The following dataset was generated:

| Author(s) | Year | Dataset title | Dataset URL | Database and Identifier |
| --- | --- | --- | --- | --- |
| Yang W | 2021 | Metagenomics genotype and phenotype association analysis on DNA modification | https://www.ncbi.nlm.nih.gov/bioproject/PRJNA714147 | NCBI Sequence Read Archive, PRJNA714147 |

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
