## [Editor Report]

This work will interest researchers who want to explore the functional potential of metagenomes. The authors present an original approach, MetaGPA, for performing enrichment analysis on cohorts of metagenomes and use it to identify a novel enzyme that can modify cytosines in DNA from natural bacteriophage populations.

---

## [Decision Letter]

**Decision letter after peer review:**

Thank you for submitting your article "A Genome-Phenome Association study in native microbiomes identifies a mechanism for cytosine modification in DNA and RNA" for consideration by *eLife*. Your article has been reviewed by 2 peer reviewers, and the evaluation has been overseen by Reviewing Editor Maria Zambrano and Michael Marletta as the Senior Editor. The reviewers have opted to remain anonymous.

The reviewers have discussed their reviews with one another, and the Reviewing Editor has drafted this letter to help you prepare a revised submission.

Essential revisions:

The work to obtain functional information from metagenomes is both novel and promising but various issued need to be addressed to support the claim of reproducibility and broader applicability. In particular, the manuscript must provide a more detailed methodology, substantially improve the characterization of their MetaGPA approach, particularly the bioinformatics, and clearly state the limitations of this method and its capacity to be used as a general tool for other searches and metagenome functional analysis.

1) Please take into account previous work, describe how these relate to the presented work and clearly indicate the motivation for MetaGPA in the introduction. The relationship between the author's program and previous attempts at GWAS-like analyses in microorganisms are not thoroughly described in the Introduction. For example, the authors do not mention TreeWAS or other similar approaches.

2) The Methods section contains information that presumably relates to the pipeline, but there is no explicit mention of MetaGPA. There are also seemingly pipeline-related methods that are not part of the codebase. The beginning section of the Results is insufficient to explain the MetaGPA method. A reproducible example is a must.

3) An attempt was made to run the code to confirm reproducibility. The GitHub link contains no information on installation or examples of how to run the code on test or new data. The code itself contains hard-coded file paths that would make it difficult, if not infeasible, to run on another person's machine. There are also dependancies that are unstated. The code must be substantially better documented to be of utility outside this study.

4) The reasons for the use of Tet2 and BGT to modify 5mC and 5hmC are unclear. The newly discovered 5-carbamoyloxymethylcytosine is unlikely to be deaminated by A3A

and hence what is the need for Tet2 and BGT? In fact, other than 5-methylcytosine, larger modifications of cytosine would not be substrates for APOBEC3A (PMID: 28472485). The investigators could treat all the DNA with A3A, divide it into two halves, and then use the USER kit to destroy DNA that contains uracils in one-half of the samples. The DNA that survives should contain a cytosine modification that protected it against A3A. Comparison of sequences of the two populations would show that only a small fraction of DNA has survived A3A+User treatment. If the ability of A3A to deaminate 5mC is a major concern, the authors do not articulate it. However, this can be easily remedied by replacing A3A with APOBEC3B-CTD or APOBEC3G-CTD neither of which is good at deaminating 5mC.

5) The USER kit contains *E. coli* Exonuclease VIII. This is problematic because this enzyme will excise oxidized pyrimidines in DNA. Thus, the DNA of any organism that routinely modifies its pyrimidines in such a way that it becomes susceptible to ExoVIII, would be eliminated from the case DNA regardless of whether it contains uracils. An AP endonuclease or simple treatment with NaOH and heat would be preferable here.

6) This methodology is designed to work for cytosine modifiers that near 100% efficient and are not sequence-specific. I do not see how this methodology could isolate genes for sequence-specific cytosine modifiers. If an enzyme strongly prefers a certain sequence motif, say 5'-TpC, then all the cytosines in the VpC sequence context (V is not T) would remain unmodified. These would be deaminated by A3A and virtually all the DNA from that organism would be destroyed by the User kit. The methodology may also have difficulty isolating genes for enzymes that modify only a fraction of the cytosine bases, say 25%. Such DNA would also be destroyed during the A3A+User treatment. Such limitations of the methodology should be carefully examined.

7) How does MetaGPA handle phylogenetic resampling (i.e., dealing with the fact that genomes are related)? This is particularly important for microorganisms. It would have been preferable to see the author's method benchmarked in a similar way, if not compared against, previous approaches to similar problems.

8) The manuscript jumps quickly into the main finding of cytosine modification. What are other applications of this technique? How could one incorporate a negative control to quantify FP rates or incorporate other controls?

9) I am somewhat confused by the discussion about the putative thymidylate synthase homologs. The authors point out that several TS homologs contain a change that is equivalent to a N177D change in *E. coli* TS. They further note that such a change in the *E. coli* enzyme results in a change of substrate from dUMP to dCMP. Does this mean that these phage genes code for dCMP methyltransferases, not thymidylate synthases? If so, they may be novel enzymes, as *E. coli* does not contain a dCMP methyltransferase and unlike TS, the normal methyl donor for cytosine methylation is S-adenosylmethionine. The methyl donor for TS is tetrahydrofolate. It would be useful to characterize these variant TS enzymes for substrate specificity and cofactor requirements.

10) It is unclear how the investigators arrived at the substrate and co-factor requirements of the newly discovered enzyme. The classic example of a carbamoyltransferase is ornithine carbamoyltransferase which tranfers the carbamoyl moiety to a nitrogen not to an oxygen. It also appears (https://www.brenda-enzymes.org/enzyme.php?ecno=2.1.3.3) that ornithine carbamoyltransferase does not require ATP as a co-factor. With that in mind, the putative new carbamoyltransferase could have modified the N4 of cytosine without the need for ATP. Did the investigators explore this possibility? Overall, a clearer rationale needs to be provided for how the requirements for the enzyme were established.

11) Interestingly, *E. coli* bacteriophage Mu carries N6-carbamoylmethyladenine. Is the enzyme that performs this reaction in any way related to the newly discovered enzyme?

12) The only justification for using phage DNA as the source of new enzymes provided by the authors is "phages are known to carry a large diversity of modifications". Another obvious reason for using phage DNA is that few phage genes contain introns. However, this convenience comes with the likely drawback that many DNA modification enzymes in cellular genomes are being missed from their screen. The authors should carefully address this issue.

13) Figure 3b is unclear. What are the units of the color (scale) bars? What am I supposed to take away from this panel? What is red and what is blue (beyond the minimal description in the figure legend)?

*Reviewer #2 (Recommendations for the authors):*

1. Page 2- Change "who is out there?" to "what is out there?".

2. Page 6- "unmodified cytosines are deaminated to uracils using the DNA cytidine

deaminase Apolipoprotein B mRNA editing enzyme catalytic polypeptide-like 3A

(APOBEC3A) (Carpenter et al., 2012)"- This is misleading on two counts. Among the APOBEC3 subfamily of cytosine deaminases, APOBEC3A is most efficient at deaminating 5mC, in addition to C. Furthermore, the ability of APOBEC3A to cause cytosine deamination was first demonstrated in HBV genome (PMID: 19169351) and the purification of the enzyme and its biochemical characterized was first reported by a different research group than cited in the manuscript (PMID: 22798497).

3. Page 11- "DNA ligase (PF14743.7, PF01068.22), and Cytidine deaminase (PF00383.24) are other domains that have been found in DNA modifying enzymes (Subramanya et al., 1996) (Bhattacharya et al., 1994)"- Did you mean DNA-cytosine deaminase? A cytidine deaminase would convert the ribonucleoside cytidine to uridine.

4. Page 18, last paragraph- The comparison of T4 genome with the contig in which carbamoyltransferase gene is found is confusing. Why is the occurrence of dCMP hydroxymethylase and β-glucosyltransferase in T4 genome "resembles" the occurrence thymidylate synthase and carbamoyltransferase in the contig? Phages have very compact genomes and tend to aggregate genes with related functions. Furthermore, dCMP hydroxymethylase is presumably an oxidase and not a transferase. If this is what led to the prediction of the reaction of the newly discovered carbamoyltransferase, it was a really inspired guess- I say that in all sincerity.

---

## [Author Response]

Essential revisions:The work to obtain functional information from metagenomes is both novel and promising but various issued need to be addressed to support the claim of reproducibility and broader applicability. In particular, the manuscript must provide a more detailed methodology, substantially improve the characterization of their MetaGPA approach, particularly the bioinformatics, and clearly state the limitations of this method and its capacity to be used as a general tool for other searches and metagenome functional analysis.

The manuscript must provide a more detailed methodology, substantially improve the characterization of their MetaGPA approach, particularly the bioinformatics:

We have significantly restructured the bioinformatic pipeline (see answer to comment 3) and provided a test dataset to benchmark it. We have also re-written the first part of the manuscript describing the MetaGPA framework (see answer to comment 2).

Clearly state the limitations of this method: we have added a entire paragraph on the limitation of MetaGPA in the revised version of the discussion:

“In these MetaGPA experiments, the sequencing libraries can be directly used as material for selection because the phenotype of interest (cytosine modification) is covalently attached to the genetic material. As such, scenarios for which the phenotype is covalently attached to the genetic material are the most straightforward applications of MetaGPA. However, for other scenarios for which phenotypes are not physically coupled with DNA or RNA, selection will have to be done while preserving the integrity of cells and viral particles to retain the link between phenotype and genotype. Thus, the limited availability of adequate selection processes that preserve such links may restrict the broader applicability of MetaGPA. While some selections can easily be adapted to MetaGPA, others may turn out to be difficult to implement.”

1) Please take into account previous work, describe how these relate to the presented work and clearly indicate the motivation for MetaGPA in the introduction. The relationship between the author's program and previous attempts at GWAS-like analyses in microorganisms are not thoroughly described in the Introduction. For example, the authors do not mention TreeWAS or other similar approaches.

The introduction of the revised manuscript contains now a paragraph that describe microbial GWAS, their achievements and limitations:

“For specific bacteria, determining the genetic basis of phenotypes has be addressed using genome-wide association studies (GWAS) [PMID: 16782339] combined with specific phylogenetic methods to account for the unique population structure of microbes [PMID: 29401456]. Examples of microbial GWAS studies have explored hundreds of isolates to identify genomic elements that are statistically associated with, for example, antibiotic resistance [PMID: 25101644], host specificity [PMID: 23818615] or virulence [PMID: 24717264]. Nonetheless, these studies are limited to known isolates and have not yet been extended to entire complex microbial communities.”

2) The Methods section contains information that presumably relates to the pipeline, but there is no explicit mention of MetaGPA. There are also seemingly pipeline-related methods that are not part of the codebase. The beginning section of the Results is insufficient to explain the MetaGPA method. A reproducible example is a must.

The reviewer brought up a great point regarding what constitutes the core MetaGPA pipeline relative to the additional analysis that we performed to validate MetaGPA. The core MetaGPA pipeline lists the protein domains that are significantly associated with the defined phenotype.

We have now clearly defined what analysis is part of the core MetaGPA pipeline and added the information to the revised version of the result section:

“The association is computed using a computational workflow composed of the core MetaGPA pipeline that defines genetic units associated with the case cohort and further analysis tools to refine these associations.”

We further explain in more detail the core MetaGPA pipeline in the first part of the manuscript.

Following the description in the revised manuscript, we have bundled the core MetaGPA pipeline into a single command line that takes both, the raw sequencing reads from the control and case group, and outputs the protein domains that are significantly associated with the defined phenotype.

We also have re-organized the Method section to reflect the above changes and created a “Computational pipeline” section to provide detailed methodology in the following sections for each step. We also rewrote the legends in Figure 1 and supplementary figure 1 to improve the explanation of the conceptual and detailed workflow.

For a reproducible pipeline example, we provided a dataset for test running at https://github.com/linyc74/MetaGPA. Please see response to question (3) for codebase related improvements.

3) An attempt was made to run the code to confirm reproducibility. The GitHub link contains no information on installation or examples of how to run the code on test or new data. The code itself contains hard-coded file paths that would make it difficult, if not infeasible, to run on another person's machine. There are also dependancies that are unstated. The code must be substantially better documented to be of utility outside this study.

We thank the reviewer’s suggestions on how to improve our code base. We have done substantial modifications such as code refactoring, packaging, and documentation, to improve the quality of our work. More specifically, we have:

1. Created a detailed, clear command-line interface with help (-h option) instructions.

2. Remove all hard-coded file paths, and make the output directory configurable in the command-line interface (-o option).

3. Make system settings (e.g. number of CPUs and max memory) configurable (-t and -m options).

4. Completely refactor the whole code base using object-oriented programming.

5. Detailed documentation in the README.md file to clearly show the following information:

a. Usage

b. Dependencies

c. Input files

d. Output files

e. Docker

6. Dockerize to completely resolve any dependency problem. Docker images could be found at https://hub.docker.com/r/linyc74/metagpa/tags.

The code is available here: https://github.com/linyc74/MetaGPA

With the aforementioned improvements, we hope future users will find it easy to apply MetaGPA to other studies.

4) The reasons for the use of Tet2 and BGT to modify 5mC and 5hmC are unclear. The newly discovered 5-carbamoyloxymethylcytosine is unlikely to be deaminated by A3Aand hence what is the need for Tet2 and BGT? In fact, other than 5-methylcytosine, larger modifications of cytosine would not be substrates for APOBEC3A (PMID: 28472485). The investigators could treat all the DNA with A3A, divide it into two halves, and then use the USER kit to destroy DNA that contains uracils in one-half of the samples. The DNA that survives should contain a cytosine modification that protected it against A3A. Comparison of sequences of the two populations would show that only a small fraction of DNA has survived A3A+User treatment. If the ability of A3A to deaminate 5mC is a major concern, the authors do not articulate it. However, this can be easily remedied by replacing A3A with APOBEC3B-CTD or APOBEC3G-CTD neither of which is good at deaminating 5mC.

The reviewer is right, larger modifications than 5mC are unlikely to be deaminated by A3A and therefore the use of Tet2 and BGT is unnecessary for the discovery of the carbamoyltransferase. This is a posteriori statement and when we designed the experiment, it was unclear whether fully C modified genomes will be present at all : we therefore took a conservative approach to include as many C modifications as possible, even the known 5mC and 5hmC. These genomes serve as a control for MetaGPA, expecting thymidylate synthases and methyltransferases to be included in the list of associated enzymes.

To clarified this point we added to the revised manuscript the underlined sentence:

“Because APOBEC3A also deaminates 5-methyl-2'-deoxycytidine (5mdC) and, to a lesser degree 5-hydroxymethyl-2'-deoxycytidine (5hmdC) (PMID: 33468551), ten-eleven translocation dioxygenase 2 (TET2) and T4 phage ß-glucosyltransferase (T4-BGT) can be used to protect 5mdC and 5hmdC prior to APOBEC3A treatment (Figure 2a). Both modifications are used as internal control for MetaGPA since the enzymes that catalyzed their formation are well characterized. These enzymes, if present in the sample, are therefore expected to be associated with modifications in our MetaGPA framework.”

Another more technical reason we used Tet2 and BGT has to do with protecting repaired DNA ends in the library construction step. Indeed, the sonication of DNA by ultrasound generates 5’ and 3’ extensions that are subsequently repaired by T4 DNA polymerase during the end-repair step (NEB Ultra II library prep kit). If regular dCTP is used, APOBEC3A will deaminate unprotected cytosines introduced in the process of end repair. To avoid this problem, we used d5mCTP instead of regular dCTP, and protected 5mC using TET2 and BGT by generating glucosylated 5hmC, which is not a substrate for APOBEC3A (se Materials and methods). As stated by the reviewer, it is possible to use APOBEC3B-CTD or APOBEC3G-CTD instead of APOBEC3A, however these two deaminases demonstrate strong substrate preference, and not commercially available (Silvas, Tania V., and Celia A. Schiffer. "APOBEC3s: DNA‐editing human cytidine deaminases." Protein Science 28, no. 9 (2019): 1552-1566). We clarified this point in the Materials and methods section:

“The dCTP was replaced with 5mdCTP in the end repair reaction (5mdCTP was used instead of regular dCTP to protect end-repaired fragments from subsequent enzymatic treatment)”.

Finally, while APOBEC3A alone will deaminate 5mC, the resulting T is not removed by USER. Instead, deaminated fragments will be sequenced but the converted sequences are uninterpretable since they are composed of only 3 nucleotides (A, T, and G).

5) The USER kit contains *E. coli* Exonuclease VIII. This is problematic because this enzyme will excise oxidized pyrimidines in DNA. Thus, the DNA of any organism that routinely modifies its pyrimidines in such a way that it becomes susceptible to ExoVIII, would be eliminated from the case DNA regardless of whether it contains uracils. An AP endonuclease or simple treatment with NaOH and heat would be preferable here.

In this work we used a mix which contains Endonuclease VIII. The reviewer is right to point out that Endonuclease VIII removes oxidative pyrimidines. However, many oxidative pyrimidines recognized by Endonuclease VIII are damaged bases including urea, 5, 6- dihydroxythymine, thymine glycol, 5-hydroxy-5- methylhydantoin, uracil glycol, 6-hydroxy-5, 6-dihydrothymine and methyltartronylurea. Many of these damaged bases are blocking damages for the amplifying polymerase and would not be sequenced even if they were not removed. Importantly, Endonuclease VIII does not remove 5hmC, which works as a handle for many modifications in phages. While AP endonuclease could be used instead of Endonuclease VIII in the USER mix, using a new USER mix (UDG + AP endonuclease) needs some optimization and is not commercially available. Treatment with NaOH is a good suggestion, however, we wanted to avoid harsh reaction conditions and heat treatment which could cause extra damage.

It would be nice, in a subsequent study, to compare contig enrichment with and without the use of Endonuclease VIII using a more promiscuous amplification polymerase. In this version of MetaGPA, we took a conservative approach with the aim to identify the subset of modifications for which the reaction steps can be easily deduced (for experimental validation). Having too many types of modified bases included in the sample cohort can confound the identification of the relevant ones.

6) This methodology is designed to work for cytosine modifiers that near 100% efficient and are not sequence-specific. I do not see how this methodology could isolate genes for sequence-specific cytosine modifiers. If an enzyme strongly prefers a certain sequence motif, say 5'-TpC, then all the cytosines in the VpC sequence context (V is not T) would remain unmodified. These would be deaminated by A3A and virtually all the DNA from that organism would be destroyed by the User kit. The methodology may also have difficulty isolating genes for enzymes that modify only a fraction of the cytosine bases, say 25%. Such DNA would also be destroyed during the A3A+User treatment. Such limitations of the methodology should be carefully examined.

The reviewer is correct: the current procedure only selects DNA for which near 100% of the cytosines are modified. The resulting enzymes are therefore more likely to show no sequence specificity which is a desired property for further applications of the enzymes. We agree with the reviewer that the scope and the rationale for targeting such non-specific enzymes has not been addressed in the manuscript. We have therefore added the following sentences to the revised manuscript:

“the experimental procedure is designed to only protect fully modified DNA, that is DNA for which cytosines are modified, irrespective of sequence contexts. A number of such organisms have previously been found, notably in bacteriophages (Kuo et al., 1968) (Revel and Georgopoulos, 1969). We hypothesize that this design would select for DNA modifying enzymes with little or no sequence specificity.”

7) How does MetaGPA handle phylogenetic resampling (i.e., dealing with the fact that genomes are related)? This is particularly important for microorganisms. It would have been preferable to see the author's method benchmarked in a similar way, if not compared against, previous approaches to similar problems.

The reviewer brought up a great point that genome sequence similarities need to be carefully considered in metagenome analysis.

First, we used metaSPAdes for metagenome assembly for which population contigs (e.i, contigs derived from multiple related species) are generated. In brief, this tool uses more aggressive settings and introduces an additional metagenomics-specific decision rule to optimize assembly within a metagenome. The detailed algorithm and benchmarking against other metagenome assemblers is published [PMID: 28298430].

We also used CD-HIT to further reduce sequence redundancy of assembled contigs. Considering the highly related sequences in microorganisms, we set the sequence identity threshold at 95%.

Finally we used RAxML which is a standard phylogenetic analytical tool [PMID: 24451623]. We enabled rapid bootstrap analysis mode and allowed for automatic determination of sufficient bootstrapping replicates (with option -f a -# autoMRE) for resampling.

8) The manuscript jumps quickly into the main finding of cytosine modification. What are other applications of this technique? How could one incorporate a negative control to quantify FP rates or incorporate other controls?

This is a multi-layered question. Here are the answer to each individual points:

The manuscript jumps quickly into the main finding of cytosine modification.

This is a good point. Before describing the application of MetaGPA on cytosine modification, we have now provided an expanded explanation of the MetaGPA pipeline and associated analysis for the reader to better understand the general concept. These expanded explanations are reflected in the manuscript main text, figure 1 and the figure 1 legend.

What are other applications of this technique?

We have a section in the discussion, describing other possible applications of MetaGPA:

“Our MetaGPA study framework is phenotype-driven and the identification of candidate genes is agnostic to prior annotations. Theoretically, it can be performed with any case/control cohort pair as long as distinct phenotypes can be partitioned through selection of the case cohort. For example, these phenotypes may include but are not limited to DNA modifications, phage sensitivity to chemicals and cell surface determinants such as O-antigen. Nonetheless, this partitioning is crucial for MetaGPA to succeed and therefore requires the development and optimization of the selection process for every new MetGPA application.”

How could one incorporate a negative control to quantify FP rates or incorporate other controls?

This is a great question raised by the reviewer. Negative and positive controls to estimate the selection procedure has been done. More specifically, we had incorporated in the samples a mixture of spiked-in genomic DNA consisting of *E. coli* (dC), XP12 (5mdC) and T4gt (5hmdC). Recovery of spiked-in modified DNAs were as expected (see manuscript).

A negative control to evaluate MetaGPA would be a microbiome for which no selection has been applied. MetaGPA requires that the control and case groups are derived from the same microbiome, thus the negative control should be a subset of the control group without selection. We therefore use the two replicate experiments done on the control group and perform a MetGPA analysis with one replicate being the control group while the other replicate being the “case” group. None of the 6737 protein domains are found significantly associated with one of the control replicates when compared to the other control replicate, indicating that the FP rate is low. We have incorporated this analysis to the revised manuscript:

“To estimate the false positive rate of association, we used our two control replicate experiments for which no selection has been done. We then perform a MetGPA analysis with one replicate being the control group and the other replicate being the “case” group. from the 6737 protein domains assessed in MetaGPA, none of the domains are found significantly associated with the case group (data not shown).”

9) I am somewhat confused by the discussion about the putative thymidylate synthase homologs. The authors point out that several TS homologs contain a change that is equivalent to a N177D change in *E. coli* TS. They further note that such a change in the *E. coli* enzyme results in a change of substrate from dUMP to dCMP. Does this mean that these phage genes code for dCMP methyltransferases, not thymidylate synthases? If so, they may be novel enzymes, as *E. coli* does not contain a dCMP methyltransferase and unlike TS, the normal methyl donor for cytosine methylation is S-adenosylmethionine. The methyl donor for TS is tetrahydrofolate. It would be useful to characterize these variant TS enzymes for substrate specificity and cofactor requirements.

We agree that the explanation in the manuscript is confusing. We have therefore added the following paragraph to the revised result section of the manuscript:

“The substrate specificity of thymidylate synthases is dictated by the residue at the position 177 (numbering relative to the *E. coli* thymidylate synthase sequence). Previous studies have demonstrated that changing this residue from asparagine to aspartate can switch the preference of a canonical thymidylate synthase from dUMP to dCMP resulting in the formation of 5mdCMP (Hardy and Nalivaika, 1992) (Graves et al., 1992) (Liu and Santi, 1992). We therefore hypothesised that position 177 should have a high differential conservation score with Asn found conserved in the unmodified contigs and Asp found conserved in modified contigs.”

More detailed information: Thymidylate synthase belongs to a family of proteins that alkylate pyrimidines at C5 by transfer of a single carbon unit from methyltetrahydrofolate (mTHF). The thymidylate synthase family encompasses four enzymatic activities: canonical thymidylate synthase (all domains of life), dUMP hydroxymethylase (Bacillus phage SPO1 and others), dCMP hydroxymethylase (bacteriophage T4 and others), and dCMP methylase (Xanthomonas phage Xp12). Sequences encoding these enzymes share a high degree of sequence and structural similarity. For example, the crystal structures of *E. coli* thymidylate synthase and it's related T4 dCMP hydroxymethylase have ~1 Å RMSD. The transfer of a carbon from mTHF to C5 results in a product intermediate that contains an exocyclic methylene (i.e. a double bond between the transferred carbon and C5 of the pyrimidine ring). For canonical exocyclic methylene can be reduced to the methyl form by hydride transfer from THF, forming dihydrofolate (DHF). But for some enzymes, the exocyclic methylene undergoes nucleophilic attack from water instead, resulting in the formation of a hydroxymethyl moiety at C5. The substrate specificity of thymidylate synthase homologs is dictated by the residue at the homologous position Asn177 (numbering relative to the *E. coli* sequence, or 221 for thymidylate synthase of *Lactobacillus* casei). For the phage T4 dCMP hydroxymethylase, this position is occupied by an aspartate residue. Researchers have demonstrated that changing this residue from asparagine to aspartate can switch the preference of a canonical thymidylate synthase from dUMP to dCMP resulting in the formation of 5mdCMP. Similarly, mutating this homologous position within the T4 dCMP hydroxymethylase from aspartate to asparagine results in an enzyme that prefers dUMP and produces 5hmdUMP. The residues governing formation of a hydroxymethyl versus a methyl group at C5 are less clear. In the case of the viral metagenome sequences encoding a DNA O-carbamoyltransferase identified here, the function of the thymidylate synthase homolog is to provide the hydroxyl "handle" for the attachment of the carbamoyl group.

References:

1. Graves, K. L., Butler, M. M., and Hardy, L. W. (1992) Roles of Cys148 and Asp179 in catalysis by deoxycytidylate hydroxymethylase from bacteriophage T4 examined by site-directed mutagenesis. Biochemistry-us. 31, 10315–10321

2. Liu, L., and Santi, D. V. (1992) Mutation of asparagine 229 to aspartate in thymidylate synthase converts the enzyme to a deoxycytidylate methylase. Biochemistry-us. 31, 5100–5104

3. Hardy, L. W., and Nalivaika, E. (1992) Asn177 in *Escherichia coli* thymidylate synthase is a major determinant of pyrimidine specificity. Proc National Acad Sci. 89, 9725–9729

4. Agarwalla, S., LaPorte, S., Liu, L., Finer-Moore, J., Stroud, R. M., and Santi, D. V. (1997) A Novel dCMP Methylase by Engineering Thymidylate Synthase †. Biochemistry-us. 36, 15909–15917

5. Liu, L., and Santi, D. V. (1993) Exclusion of 2’-deoxycytidine 5’-monophosphate by asparagine 229 of thymidylate synthase. Biochemistry-us. 32, 9263–9267

10) It is unclear how the investigators arrived at the substrate and co-factor requirements of the newly discovered enzyme. The classic example of a carbamoyltransferase is ornithine carbamoyltransferase which tranfers the carbamoyl moiety to a nitrogen not to an oxygen. It also appears (https://www.brenda-enzymes.org/enzyme.php?ecno=2.1.3.3) that ornithine carbamoyltransferase does not require ATP as a co-factor. With that in mind, the putative new carbamoyltransferase could have modified the N4 of cytosine without the need for ATP. Did the investigators explore this possibility? Overall, a clearer rationale needs to be provided for how the requirements for the enzyme were established.

Those are excellent points. The enzyme requires 3 substrates: ATP, carbamoyl phosphate and 5hmdC. We have explained throughout the manuscript the reason for 5hmdC (see also answer to reviewer 2 point 4). The choice of ATP and carbamoyl phosphate substrates were guided by the enzymatic characterization of TobZ previously published by Parthier et al., 2012. In this publication, the authors used ATP and carbamoyl phosphate as substrates for TobZ, a member of the O-carbamoyltransferase that has been enzymatically and structurally characterized and has strong homology with our enzyme. We have clarified this point in the revised manuscript (underlined text):

“The predicted reaction was tested by enzymatic assays and results showed that every substrate, namely carbamoyl phosphate, ATP, 5hmdC (genomic T4gt DNA was used as substrate in these experiments) and the enzyme were indispensable for the reaction (Supplementary Figure 5c-d). The choice of carbamoyl phosphate and ATP substrates were guided by the enzymatic characterization of TobZ previously published (Parthier et al., 2012).”

Regarding the requirement of ATP, we have performed the reaction with and without ATP. The result showed that ATP is a required cofactor (see Supplementary Figure 5c).

Finally regarding the possibility of a reaction on the N4 of cytosine without the need for ATP, we found that neither the predicted mass of an N-carbamoylated 5hmC, nor an N-carbamoylated cytidine matched the modified base produced in our reconstituted in vitro reaction.

11) Interestingly, *E. coli* bacteriophage Mu carries N6-carbamoylmethyladenine. Is the enzyme that performs this reaction in any way related to the newly discovered enzyme?

Following the reviewer’s comment, we explored whether N6-carbamoylmethyladenine in Mu phage (Methylcarbamoylase mom) is related to carbamyltranferase.

Methylcarbamoylase mom protein belongs to the GCN5-related N-acetyltranferase (GNAT) family. It is suggested that the Mom enzyme uses a coenzyme-A carrier to donate a formamide moiety to an m6A substrate. This is different from a carbamoyltranferase. We did protein sequence comparison with protein BLAST and HMMER. The results showed no similarity between the enzyme we found and Methylcarbamoylase mom (uniprot p06018). HMMER search against the pfam database (Pfam-A.hmm) for the two protein sequences has no overlapping pfam hit. Together, these results indicate that these two enzymes are mechanistically and evolutionarily unrelated.

12) The only justification for using phage DNA as the source of new enzymes provided by the authors is "phages are known to carry a large diversity of modifications". Another obvious reason for using phage DNA is that few phage genes contain introns. However, this convenience comes with the likely drawback that many DNA modification enzymes in cellular genomes are being missed from their screen. The authors should carefully address this issue.

The reviewer is right that we did not carefully explain the underlying reasons for having the current design of MetaGPA enriching for phages. Indeed, phages are known to carry a large diversity of modifications but importantly (and we have omitted to state this point in the manuscript) phages tend to fully modify their genome to evade any restriction systems. Other organisms tend to have modifications only in certain sequence contexts. Because our experimental procedure is designed to only select for fully modified cytosines (also addressed in essential revision 6), we hypothesized that the phage fraction would be the most likely fraction of the microbiome to contain such DNA.

We have included this point in the revised manuscript (see underlined text):

“We aim at obtaining the phage-enriched fraction of each microbiome. The rationale for this selection is based on the fact that phages are known to carry a large diversity of modifications (Weigele and Raleigh, 2016) and phages have been shown to fully modify their genomes irrespectively of sequencing context which is a prerequisite for our MetaGPA selection.”

13) Figure 3b is unclear. What are the units of the color (scale) bars? What am I supposed to take away from this panel? What is red and what is blue (beyond the minimal description in the figure legend)?

We realized Figure 3b is less informative for the point we want to convey. We have now redone the kmer analysis, this time annotating each kmer in one of the 3 following categories: [1] kmer found in the modified contigs (in orange), [2] kmer found in the unmodified contigs (in blue) and [3] kmer that did not result in assembled contigs (grey). In the revised manuscript, we replace figure 3b accordingly and added the new legend:

“b, Normalized frequency of k-mer in sequencing reads from control (x-axis) compared to case (y-axis) groups. Each dot represents a unique 16-mer and is colored according to the resulting contig category it belonged to. Orange: kmer at modified contigs; blue: kmer at unmodified contigs; grey: unassembled kmers. See c below for definition of modified/unmodified contigs. Subsamples of randomly selected 0.1% of all possible kmers were used for plotting.”

Reviewer #2 (Recommendations for the authors):1. Page 2- Change "who is out there?" to "what is out there?".

We have now changed this sentence to “what is out there?”

2. Page 6- "unmodified cytosines are deaminated to uracils using the DNA cytidinedeaminase Apolipoprotein B mRNA editing enzyme catalytic polypeptide-like 3A(APOBEC3A) (Carpenter et al., 2012)"- This is misleading on two counts. Among the APOBEC3 subfamily of cytosine deaminases, APOBEC3A is most efficient at deaminating 5mC, in addition to C. Furthermore, the ability of APOBEC3A to cause cytosine deamination was first demonstrated in HBV genome (PMID: 19169351) and the purification of the enzyme and its biochemical characterized was first reported by a different research group than cited in the manuscript (PMID: 22798497).

We have clarified the fact that APOBEC3A also deaminates 5mC in the revised manuscript (underlining text) and added the two references:

“unmodified cytosines are deaminated to uracils using the DNA cytidine deaminase Apolipoprotein B mRNA editing enzyme catalytic polypeptide-like 3A (APOBEC3A) (PMID: 19169351, PMID: 22798497, Carpenter et al., 2012), and subsequently excised by Uracil-Specific Excision Reagent (USER) (Bitinaite et al., 2007), resulting in fragmented DNA. Because APOBEC3A also deaminates 5-methyl-2'-deoxycytidine (5mdC) and, to a lesser degree 5-hydroxymethyl-2'-deoxycytidine (5hmdC) (PMID: 33468551), ten-eleven translocation dioxygenase 2 (TET2) and T4 phage ß-glucosyltransferase (T4-BGT) can be used to protect 5mdC and 5hmdC prior to APOBEC3A treatment (Figure 2a)”

3. Page 11- "DNA ligase (PF14743.7, PF01068.22), and Cytidine deaminase (PF00383.24) are other domains that have been found in DNA modifying enzymes (Subramanya et al., 1996) (Bhattacharya et al., 1994)"- Did you mean DNA-cytosine deaminase? A cytidine deaminase would convert the ribonucleoside cytidine to uridine.

We would like to thank the reviewer for spotting this typo. We meant the protein family PF00383.24 (cytidine and deoxycytidylate deaminase zinc-binding region). We corrected it in the manuscript: For example, a subset of enzymes containing the thymidylate synthase domain (PF00303.20) have been shown to produce hydroxymethylpyrimidines (Neuhard et al., 1980). DNA ligase (PF14743.7, PF01068.22), and Cytidine/deoxycytidylate deaminase (PF00383.24) are other domains that have been found in DNA modifying enzymes (Subramanya et al., 1996) (Bhattacharya et al., 1994).

4. Page 18, last paragraph- The comparison of T4 genome with the contig in which carbamoyltransferase gene is found is confusing. Why is the occurrence of dCMP hydroxymethylase and β-glucosyltransferase in T4 genome "resembles" the occurrence thymidylate synthase and carbamoyltransferase in the contig? Phages have very compact genomes and tend to aggregate genes with related functions. Furthermore, dCMP hydroxymethylase is presumably an oxidase and not a transferase. If this is what led to the prediction of the reaction of the newly discovered carbamoyltransferase, it was a really inspired guess- I say that in all sincerity.

We have not properly explained how the T4 phage gene configuration led us to hypothesize the mechanism of action of the carbamoyltransferase. We have therefore added to the revised version of the manuscript the following paragraph:

“The co-occurrence of carbamoyltransferase and thymidylate synthase homologs specifically in modified contigs (Figure 4c) resembles the arrangement of the T4 phages for which genes coding for the dCMP hydroxymethylase and β-glucosyltransferase co-occur on the genome (Miller et al., 2003). The T4 dCMP hydroxymethylase is homologous to thymidylate synthase and transfers a carbon from methyltetrahydrofolate (mTHF) to C5 of the pyrimidine ring producing an exocyclic methylene in the active site of the enzyme (Graves et al. 1992). However, unlike thymidylate synthase, the methylene intermediate undergoes nucelophilic attack by water producing a hydroxymethyl group. Following incorporation of 5hmC into DNA during replication, T4 β-glucosyltransferase transfers a glucose to the hydroxyl moiety of 5hmC. Thus, the pairing of a carbamoyltransferase with dCMP hydroxymethylase led us to hypothesize a novel form of DNA modification, in which the carbamoyltransferase catalyzes the transfer of a carbamoyl group to the nucleophilic hydroxyl acceptor group of 5hmdC producing 5-carbamoyloxymethylcytosine (5cmdC) (Figure 6a).”